



# SCOPE 2.0: A model to simulate vegetated land surface fluxes and satellite signals

Peiqi Yang[1], Egor Prikaziuk[1], Wout Verhoef[1], and Christiaan van der Tol[1]

[1]University of Twente, Faculty ITC, P.O. Box 217, 7500 AE Enschede, The Netherlands

**Correspondence:** Christiaan van der Tol (c.vandertol@utwente.nl)

**Abstract.** The Soil Canopy Observation of Photosynthesis and Energy fluxes (SCOPE) model aims at linking satellite observations in the visible, infrared and thermal domains with land surface processes in a physically based manner, and quantifying the micro-climate in the canopy. It simulates radiative transfer in the soil, leaves and vegetation canopies, as well as photosynthesis and non-radiative heat dissipation through convection and mechanical turbulence. Since the first publication 11 years ago, SCOPE has been applied in remote sensing studies of solar-induced chlorophyll fluorescence (SIF), energy balance fluxes, gross primary productivity (GPP) and directional thermal signals. Here we present a thoroughly revised version, SCOPE 2.0, which features a number of new elements: (1) It enables the definition of layers consisting of leaves with different properties, thus enabling the simulation of vegetation with an understory or with a vertical gradient in leaf chlorophyll concentration; (2) It enables the simulation of soil reflectance; (3) It includes the simulation of leaf and canopy reflectance changes induced by the xanthophyll cycle; and (4) The computation speed has been reduced by 90% compared to earlier versions due to a fundamental optimization of the model. These new features improve the capability of the model to represent complex canopies and to explore the response of remote sensing signals to vegetation physiology. The improvements in the computational efficiency make it possible to use SCOPE 2.0 routinely for the simulation of satellite data and land surface fluxes. It also strengthens the operability for numerical retrieval of land surface products from satellite or airborne data.

## 1 Introduction

Vegetation, as a dynamic component of the earth system, affects the climate via its influence on the exchange of energy and matter between the land surface and the atmosphere. Quantification of this exchange is relevant for a wide range of applications including weather prediction, climate projections, agriculture, and ecological and hydrological studies.

Earth observation with satellites can be used to monitor key characteristics of vegetation that are responsible for the surface-atmosphere exchanges, and identify changes therein. The most commonly used remote sensing indicator of vegetation biophysical and biochemical properties is reflectance (Ollinger, 2011). For example, the MODIS (Moderate Resolution Imaging Spectroradiometer) normalized vegetation reflectance index (NDVI), and the MERIS (MEdium Resolution Imaging Spec-





trometer) Terrestrial Chlorophyll Index (MTCI) have been empirically correlated with canopy leaf area index (LAI) and leaf
chlorophyll content, respectively (Huete et al., 2002). Recent developments of sun-induced chlorophyll fluorescence (SIF)
have offered an additional way to monitor vegetation (Mohammed et al., 2019). SIF has been successfully used to estimate
gross primary production (GPP) (Guanter et al., 2014; Ryu et al., 2019) and stress detection (Ač et al., 2015; Rossini et al.,
2015). Additional to reflectance and SIF, thermal signals provide insights into the physical processes of surface energy and
water balance, such as demonstrated by the mapping of evapotranspiration globally with satellite thermal radiance (Nemani
and Running, 1989; Allen et al., 2007).

Combined radiative transfer and plant physiological modelling is a promising way to investigate the exchanges of energy,
water and carbon among soil, vegetation and atmosphere, and to develop remote sensing techniques for monitoring of vegeta-
tion functioning. Many factors affect the signals observed from remote sensing, including sun-observation geometry, vegetation
canopy structure and composition of the Earth's surface and atmosphere. Physically consistent exploitation of remote sensing
data, therefore, requires the modelling of radiative transfer in the soil-vegetation-atmosphere system. Radiative transfer models
(RTMs) describe the relationship between vegetation characteristics and remote sensing observations obtained under varying
sun-observer geometry. However, for a complete understanding of the role of vegetation in the energy budget of the Earth's
surface, radiative transfer modelling is not sufficient. One also needs to model non-radiative processes of energy dissipation via
photosynthesis, phase transitions of water, heat storage and turbulent heat exchange between the surface and the atmosphere.
This enables investigations beyond the monitoring of vegetation biophysical and biochemical properties, towards monitoring
of fluxes.

The Soil-Canopy-Observation of Photosynthesis and Energy fluxes (SCOPE) model simulates the radiative transfer of in-
cident light and thermal and fluorescence radiation emitted by soil and plants, component temperatures, photosynthesis and
turbulent heat exchange (Van der Tol et al., 2009). In SCOPE, the radiative transfer and the non-radiative energy fluxes are
computed in an assemblage of leaves and soil. The energy balance is maintained at all levels of spatial aggregation. Maintain-
ing an energy budget is necessary for the simulation of thermal radiation, which depends on the within-canopy temperature
distribution. To obtain this distribution, stomatal aperture, latent and sensible heat fluxes of individual elements have to be
resolved together with the radiative fluxes in the vegetation canopy.

SCOPE has been applied in a wide range of studies. Thanks to the coupling of photosynthesis and radiative transfer of
fluorescence in the SCOPE model, it has been used as a convenient tool for in-depth process-based studies to unravel the
relationship between fluorescence and photosynthesis (Damm et al., 2015; Verrelst et al., 2016; Migliavacca et al., 2017).
Besides, it has also been used for simulating directional anisotropy of satellite-measured surface temperatures (Duffour et al.,
2015), for predicting evapotranspiration (Galleguillos et al., 2011), and as a benchmark for other simple radiative transfer
models (Bian et al., 2020). Contemporary simulations of satellite observations and plant physiological processes make SCOPE
a useful tool to monitor dynamic vegetation response to environmental conditions (Zhang et al., 2014; Pacheco-Labrador et al.,
2019).

Since the original publication, SCOPE been extended with new features:

1. The soil reflectance model BSM (Verhoef et al., 2018; Yang et al., 2020b) has been introduced.





2. The radiative transfer of fluorescence has been improved (Van der Tol et al., 2019b).

3. Changes in reflectance due to xanthophyll pigment changes have been included (Vilfan et al., 2018).

4. The RTMs in the SCOPE have been adapted for multi-layer canopies (Yang et al., 2017).

These new features have not been described together. In the new version of SCOPE (SCOPE 2.0) presented here, these improvements are coherently incorporated. Moreover, the model has been optimized in many ways to improve the computational efficiency, and the options to provide data input formats have been extended. We present a description of the basic functionality
of the model followed by several recent developments.

## 2   General description of SCOPE

### 2.1   Starting points

SCOPE is designed to simulate photosynthetic, hydrological, and radiative transfer processes at the vegetated land surface. For these purposes, it combines several RTMs with a leaf biochemical model and an aerodynamic resistance scheme. These
models provide simulations of emanating hyperspectral radiance and net radiation $R_n$ (via radiative transfer processes), photosynthesis rates (via photosynthetic processes), and sensible heat flux $H$, latent heat flux $\lambda E$ and ground heat flux $G$ (via micro-meteorological processes), for both individual elements of the land surface (e.g., soil and leaves) and the whole vegetation stand. In order to meet the requirements of broad applicability, the models are as much as possible physically based. The model has been developed further since the first publication Van der Tol et al. (2009), but the main model structure and
functionality remain. The SCOPE model described in this section refers to the first-release version in 2009.

The central idea of SCOPE is the modelling of interactions between radiative and non-radiative fluxes among elements of the vegetation canopy. Remote sensing signals, such as reflectance, fluorescence and thermal radiance, are outputs of these interactions. The modelling of radiative fluxes facilitates the simulation of optical properties (i.e., reflectance, transmittance and absorptance) of soil, leaves and canopies. This is complemented with the modelling of non-radiative fluxes in vegetation
canopies, respecting energy conservation at all levels of spatial aggregation from the photosystem to the whole stand. The energy budget is determined by both the radiative transfer of incident and emitted (thermal) radiation, and the exchange of (latent) heat with the atmosphere. The surface temperature is resolved as the outcome of this balance.

### 2.2   Model domain and representation

In the spatial domain, the typical representation of land surfaces in SCOPE is a vegetation layer consisting of leaves bounded
underneath by a soil surface. The representation of the vegetation layer is one-dimensional in the sense that fluxes in the vertical ($z$) direction are considered only. This implies that even if the model is applied pixel-by-pixel in a spatial grid, the horizontal interactions between neighbouring pixels are not considered. Thus, typical 3-D effects in the vegetation, such as boundary





effects at the edge of fields or forests, or effects of topography and horizontal heat advection are not included. Radiative transfer is based on turbid medium representations of both leaves and canopies.

In the temporal domain, SCOPE assumes steady-state conditions. This means that a simulation with SCOPE outputs the energy and spectrally resolved radiation budgets of the surface for a single set of surface and weather characteristics at one moment in time. The lack of memory of state variables in time also means that storage of carbon and water are not considered, and similarly, vegetation growth is not simulated. A complete run of SCOPE may consist of many simulations, either for one location as a function of time, or for different locations or surface types, but the simulations in such a sequence are treated independently without interactions, and thus the order of model simulations is arbitrary. One exception is the (optional) modelling of the soil heat budget with a thermal inertia approach, which is described in section 3.4.

In the spectral domain, SCOPE simulates visible to thermal infrared radiance from 0.4 to 50$\mu$m within and above vegetation canopies. The spectral resolutions in the spectral regions from 0.4 to 2.5$\mu$m, from 2.5 to 15$\mu$m, and from 15 to 50 $\mu$m are 1 nm, 100 nm, and 1000 nm, respectively. It also covers the fluorescence emission spectral region from 640 to 850 nm with a resolution of 1 nm. It is noted that the spectral resolutions in these regions are easily adapted to simulation requirements and spectral input data.

## 2.3 Structure of the model

The model code at the highest hierarchical level, SCOPE, calls sub-models which operate in series. The main sub-models are listed in Table 1. Besides the listed sub-models for radiative transfer and energy balance, SCOPE requires functions for input, output and some supporting functions (such as Planck's equation). Therefore, all the functions used in SCOPE can be organized in four types: (1) RTMs, (2) modules for energy balance, (3) input-output functions, and (4) supporting functions.

### 2.3.1 RTMs

SCOPE includes seven RTMs, which together simulate the spectrally resolved radiance emanating from the vegetation: one for the soil (BSM, only available in SCOPE 2.0, Verhoef et al., 2018; Yang et al., 2020b), one for the leaf (Fluspect, Vilfan et al., 2016, 2018), and five for the whole stand, i.e. the combined system of soil and foliage. They include one RTM for incident radiation from the sun and the sky (RTMo), two for thermal radiation emitted by the soil and vegetation (RTMt_sb and RTMt_planck), one for chlorophyll fluorescence, RTMf (Van der Tol et al., 2009, 2019b), and one for the dynamic modulations of leaf reflectance and transmittance due to pigment changes in the xanthophyll cycle (RTMz, only available in SCOPE v1.70 or later, Vilfan et al., 2018).

Four types of fluxes are involved in the radiative transfer processes, namely a direct solar flux, two hemispherical (semi-isotropic) diffuse fluxes (up- and downward) and a flux in the direction of viewing. Following the Kubelka-Munk theory, the radiative transfer in the vertical direction is expressed with a set of linear differential equations (Verhoef, 1984). These equations are solved either with analytical or numerical approaches.





**Table 1.** Main sub-models in SCOPE.

| sub-models | main functions | main input | main output |
|---|---|---|---|
| BSM | simulating soil reflectance | soil moisture, brightness and two spectral shape related parameters | anisotropic soil reflectance |
| Fluspect | leaf RTM | leaf biophysical properties | leaf reflectance, transmittance and fluorescence emission matrices |
| RTMo | RTM for incident radiation | canopy structure, leaf reflectance, transmittance and soil reflectance | canopy reflectance, radiation absorbed by each leaf |
| RTMf | RTM for fluorescence fluxes | canopy structure, leaf reflectance, transmittance, soil reflectance and fluorescence emission matrices | fluorescence of each leaf and of the whole canopy |
| RTMt_sb/RTMt_planck | RTM for thermal fluxes | leaf temperature, incoming thermal radiation, emissivity of soil and leaves | thermal emission of each leaf and of the whole canopy |
| RTMz | RTM for fluxes induced by the xanthophyll cycle | leaf absorbed radiation, canopy structure, leaf reflectance, transmittance, soil reflectance | dynamic modulations of canopy reflectance |
| biochemical | biochemical model for photosystem energy partitioning | leaf absorbed radiation, leaf temperature, photosynthetic parameters | photosynthesis rate, fluorescence emission efficiency and heat dissipation |
| ebal | energy balance module | leaf absorbed radiation, leaf temperature | sensible and latent heat fluxes |

### 2.3.2 Energy balance module

The energy balance module in SCOPE minimizes the energy balance closure error $e_{ebal}$:

$$e_{ebal} = R_n - H + \lambda E - G \qquad (1)$$

for all leaf and soil elements by iteratively updating their temperature. In this equation, $R_n$ is the net radiation, $H$ the sensible heat flux, $\lambda E$ the latent heat flux, and $G$ the ground heat flux (i.e., zero for leaf elements), all in $\mathrm{Wm}^{-2}$. In the energy balance, chemical conversions (e.g., photosynthesis and respiration) and fluorescence are neglected.

The net radiation is obtained after spectral integration of the radiative transfer modules for incident radiation (RTMo) and internally generated thermal radiation (RTMt). The radiative transfer for incident radiation is computed before the energy balance closure loop, while the internally generated thermal radiation is calculated within this loop, because of its dependence on leaf and soil temperatures. The sensible and latent heat fluxes are calculated with an aerodynamic resistance scheme, where resistances for the leaf and soil boundary layer, the vegetation layer and the atmospheric surface layer apply. The aerodynamic

resistances depend on atmospheric stability, wind speed and surface roughness (Van der Tol et al., 2009). For latent heat flux of leaves, a stomatal resistance is calculated with a combined photosynthesis and stomatal model (Van der Tol et al., 2014), while for the soil, a surface resistance applies which is either a pre-defined input, or an empirical function of soil moisture.





### 2.3.3 Leaf biochemical model

The biochemical model simulates the energy partitioning into fluorescence, heat or photochemistry in photosystems (Maxwell
and Johnson, 2000; Van der Tol et al., 2014). It is based a conventional photosynthesis model of Collatz et al. (1991) for C3
and Collatz et al. (1992) for C4 vegetation, in which photosynthetic rates (or photosynthetic light use efficiency) are simulated
as a function of leaf temperature, ambient radiation levels, intercellular $CO_2$ concentrations $C_i$, and other leaf physiological
parameters (e.g., photosynthetic pathways, maximum carboxylation rate $V_{cmo}$).

Van der Tol et al. (2014) established empirical relationships between fluorescence emission efficiency and photosynthetic
light use efficiency under various environmental conditions by using active fluorescence measurements. With these relation-
ships, the fraction of the absorbed radiation by a leaf emitted as fluorescence and dissipated as heat can be simulated.

### 2.3.4 Interactions among the sub-models

Fig. 1 is a schematic overview of the SCOPE model structure, which also shows the connections among the sub-models. A full
list of input parameters is provided in Table 2. A simulation with SCOPE starts with calculating soil reflectance (BSM), the
leaf reflectance and transmittance and fluorescence emission excitation matrices (Fluspect). These simulations of soil and leaf
optical properties, together with canopy structure and irradiance, are the input of canopy RTMs. The sub-model for radiative
transfer of solar and sky radiation (RTMo) takes leaf optical properties, and soil reflectance as input and simulates canopy
reflectance and radiation fluxes including the net absorbed solar radiation by soil and leaves. RTMf takes the leaf fluorescence
emission excitation matrices and the radiation fluxes as input, and simulates canopy fluorescence.

The radiative transfer of emitted thermal radiation relies on the temperatures of soil and leaves, which are not known a priori.
For this reason, the thermal radiative transfer model is carried out in the energy balance closure loop, as described in section
2.3.2. For the purpose of computational efficiency, the radiative transfer of emitted thermal radiation is carried out in broadband
by using RTMt_sb. The letters 'sb' denote the use of the Stefan–Boltzmann law to describe the spectrally integrated radiance
from a leaf or soil in terms of its temperature. Leaf temperature is also used together with the radiation absorbed by leaf
chlorophyll pigments and other leaf physiological parameters, to simulate photosystem energy partitioning in the biochemical
model (Van der Tol et al., 2014). The energy balance residual is used to update the initial estimate of the temperature of each
element.

After energy balance closure, the thermal radiation fluxes are simulated spectrally resolved in observation direction by using
RTMt_planck, where 'planck' denotes the use of Planck's law to describe the spectrally resolved radiance from a leaf or
soil in terms of its temperature. The radiative transfer of the emitted fluorescence is simulated with RTMf. This module uses
the radiative fluxes interacting with leaves as simulated with RTMo, and the fluorescence emission matrices simulated with
Fluspect, to simulate leaves' fluorescence emission, which is aggregated to canopy fluorescence signals. Finally, the effect of
(small) changes in reflectance and transmittance due to the illumination and temperature dependent xanthophyll epoxidation
state are simulated with RTMz.





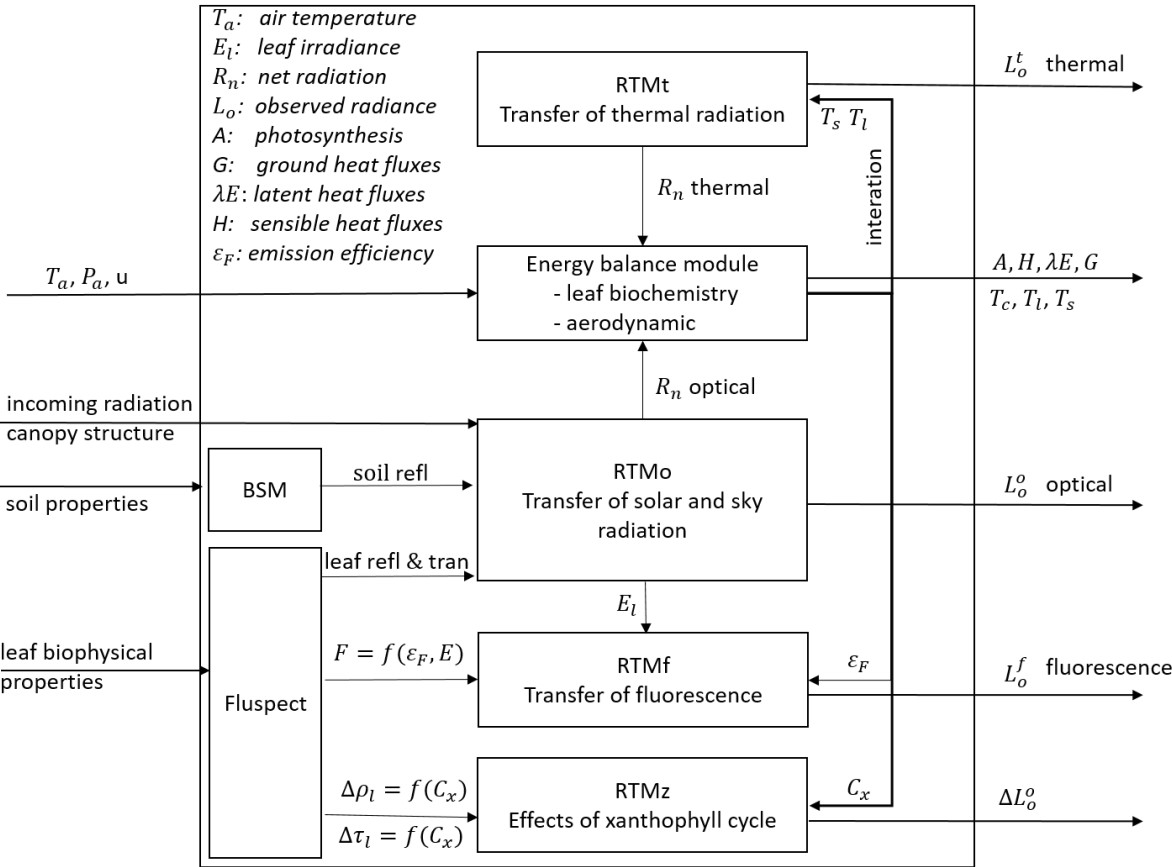

**Figure 1.** Schematic overview of the SCOPE model structure. For a complete list of input data, see Table 2

## 2.4 Model inputs and outputs

### 2.4.1 Input variables for soil, vegetation and meteorology

The inputs of the SCOPE model comprise soil, leaf and canopy properties as well as sun-observer geometry and meteorological conditions. Besides the intermediate variables, which are passed between the sub-models, the main input variables of SCOPE are given in Table 2.

Leaf biophysical and biochemical parameters characterize leaf pigment, water and dry matter contents, which determine leaf optical properties. Canopy structural parameters describe the arrangement of the leaves in the canopy. Sun-observer geometry is determined by sun and observer's zenith angles and their absolute azimuth difference. Both the canopy structural parameters and sun-observer geometry strongly affect remote sensing signals observed above the canopy. It is worth noting that viewing angles only affect the directional signals observed at the top of a canopy, such as reflectance, fluorescence and thermal radiation,





but they have no impact on the radiative fluxes inside the canopy, canopy photosynthesis and respiration and non-radiative fluxes, such as latent and sensible heat.

The meteorological inputs for SCOPE include the typical synoptic weather variables of air temperature, humidity, wind speed, air pressure and the concentrations of oxygen and carbon dioxide. All these inputs are required close to the Earth surface. The height above the surface of the terrain for which they are specified must be given in the input, as input $z$, typically

2.5 times the vegetation height. Thus, $z$ is not the height of the terrain above sea level, but rather the height above the terrain at in surface layer, where the wind profile is logarithmic. The value of $z$ must be given in the input as it is used to calculate the aerodynamic roughness of the surface.

### 2.4.2 Input irradiance for the atmosphere boundary condition

In addition to the variables listed in Table 2, SCOPE requires the radiative properties of the atmosphere as the upper boundary

condition. They can be provided in two different ways. The first option is to provide irradiance at the bottom of atmosphere (BOA) in the form of a file with two columns representing the spectra of direct solar irradiance $E_{sun}$, and diffuse sky irradiance $E_{sky}$ [Wm$^{-2}\mu$m$^{-1}$sr$^{-1}$], respectively. These spectra could either be measured in the field or generated with an atmospheric RTM (e.g. MODTRAN,  Berk et al., 1999). Using an atmospheric RTM has the disadvantage that $E_{sky}$ may not be accurate, because $E_{sky}$ also depends on the surface (canopy) reflectance in the surroundings, which may not be known a priori in the

atmospheric radiative transfer simulation. Therefore, if the surface reflectance assumed in the atmospheric radiative transfer simulation largely differs from the canopy reflectance produced by SCOPE, errors in $E_{sky}$ occur. The second and preferred option is using an atmospheric RTM to generate some optical properties of the atmosphere rather than the direct output of $E_{sky}$ and $E_{sun}$. The optical properties should include the following coefficients.

- $E_s \cos(\theta_s)$, the product of the solar irradiance at TOA and the cosine of the solar zenith angle. This product is the

irradiance at TOA projected on the surface [Wm$^{-2}\mu$m$^{-1}$sr$^{-1}$].

- $\rho_{dd}$, the diffuse reflectance of the atmosphere (i.e. the spherical albedo).

- $\tau_{ss}$, the direct atmospheric transmittance in the direction of the sun.

- $\tau_{sd}$, the diffuse atmospheric transmittance for solar incidence.

- $L_a$, the thermal emission by atmosphere at BOA, towards the surface (Wm$^{-2}\mu$m$^{-1}$sr$^{-1}$).

The coefficients listed can be extracted from MODTRAN simulations by using the T18 system, which is described in detail in Verhoef et al. (2018). A database of the optical coefficients for several typical atmospheric conditions is provided together with the SCOPE model. With these coefficients, SCOPE can simulate the BOA direct and diffuse irradiance spectra in the module RTMo with consideration of surface-atmosphere interactions. The BOA irradiances $E_{sun}$ and $E_{sky}$ are calculated in





**Table 2.** Main input variables of SCOPE

| symbol | abbreviation | unit | default value | sub-model | description |
|---|---|---|---|---|---|
| $C_{ab}$ | Cab | [$\mu$gcm$^{-2}$] | 80 | Fluspect | leaf chlorophyll concentration |
| $C_{ca}$ | Cca | [$\mu$gcm$^{-2}$] | 20 | Fluspect | leaf carotenoid concentration |
| $C_w$ | Cw | [cm] | 0.01 | Fluspect | equivalent water thickness in leaves |
| $C_s$ | Cs | [] | 0 | Fluspect | leaf senescence parameters |
| $C_{dm}$ | Cdm | [g cm$^{-2}$] | 0.012 | Fluspect | leaf dry matter content |
| $C_{ant}$ | Cant | [$\mu$g cm$^{-2}$] | 0 | Fluspect | Anthocyanin content |
| $N$ | N | [] | 1.4 | Fluspect | leaf structure parameter |
| $L$ | LAI | [] | 3 | canopy RTMs | projected leaf area per unit ground area |
| $h_c$ | hc | [m] | 2 | canopy RTMs | vegetation height |
| $LIDFa$ | LIDFa | [] | -0.35 | canopy RTMs | parameter for the mean leaf zenith angle |
| $LIDFb$ | LIDFb | [] | -0.15 | canopy RTMs | bimodality of leaf angle distribution |
| $\theta_s$ | tts | [deg] | 30 | canopy RTMs | solar zenith angle |
| $\theta_o$ | tto | [deg] | 0 | canopy RTMs | viewing zenith angle |
| $\psi$ | psi | [deg] | 0 | canopy RTMs | absolute azimuth difference |
| $R_{si}$ | Rin | [Wm$^{-2}$] | 600 | canopy RTMs | shortwave irradiance |
| $R_{li}$ | Rli | [Wm$^{-2}$] | 300 | canopy RTMs | longwave irradiance |
| $p_a$ | p | [hPa] | 970 | energy balance | air pressure |
| $T_a$ | T | [°C] | 20 | energy balance | air temperature |
| $u$ | u | [ms$^{-1}$] | 2 | energy balance | wind speed |
| $e_a$ | ea | [hPa] | 15 | energy balance | vapour pressure |
| $z$ | z | 2 | [m] | energy balance | measurement height |
| $\Theta$ | SMC | [] | 25 | BSM, energy balance | surface volumetric soil moisture content |
| $B$ | BSMBrightness | [] | 0.5 | BSM | soil brightness |
| $\varphi$ | BSMlat | [deg] | 25 | BSM | soil 'latitude' parameter (not geographical) |
| $\lambda$ | BSMlon | [deg] | 45 | BSM | soil 'longitude' parameter (not geographical) |
| $C_a$ | Ca | [ppm] | 380 | biochemical model | atmospheric $CO_2$ concentration |
| $V_{cmo}$ | Vcmo | [$\mu$mol m$^{-2}$] | 70 | biochemical model | carboxylation capacity at $25 \deg C$ |
| $m$ | m | [] | 12 | biochemical model | Ball-Berry stomatal parameter |





the function RTMo as:

$$E_{sun} = E_s \cos(\theta_s)\overline{\tau_{ss}}$$

$$E_{sky} = \frac{E_s \cos(\theta_s)(\overline{\tau_{sd}} + \overline{\tau_{ss}\rho_{dd}}r_{sd}) + \pi((1 - r_{dd})L_s\overline{\rho_{dd}} + L_a)}{1 - \overline{\rho_{dd}}r_{dd}} \qquad (2)$$

where $r_{sd}$ and $r_{dd}$ are the surface reflectance for direct and diffuse incoming radiation, respectively and $L_s$ the thermal emission by the (vegetated) surface ($\mathrm{Wm^{-2}\mu m^{-1}sr^{-1}}$). All of them are simulated with SCOPE. The overbars denote the spectral averaging to the SCOPE resolution (1 nm in the VNIR). Note that $\tau_{ss}$ and $\rho_{dd}$ are aggregated to the SCOPE resolution separately, but also the product $\tau_{ss}\rho_{dd}$, in order to accommodate spectral correlation effects in the finite bands. The coupling with
the atmosphere is descried in detail in Verhoef et al. (2018) and Yang et al. (2020b).

Finally, SCOPE offers the possibility to provide additional values for the spectrally integrated irradiance (direct solar radiation $E_{sun}$ plus $E_{sky}$) over the ranges from 0.4 to 2.5$\mu$m, and 2.5 to 50$\mu$m. These are the input fields $R_{si}$ and $R_{li}$, respectively. However, it is not necessary to specify these inputs, because the broadband irradiances $R_{si}$ and $R_{li}$ are already calculated internally as the integral of the irradiance spectra. If the values for these two inputs are specified, then the solar and sky ir-
radiance spectra $E_{sun}$ plus $E_{sky}$ are linearly scaled (each by the same factor so that the ratio $E_{sun}/E_{sky}$ remains unaltered) in the two spectral regions separately, to match the values provided for $R_{si}$ and $R_{li}$. This option can be useful if time series of synoptic weather data are used as input, and if it is computationally not feasible to carry out atmospheric radiative transfer simulations for every time step separately. For coupled surface-atmosphere simulations this is not recommended, because of obvious inconsistencies between SCOPE and the atmospheric model. In that case, the input fields for $R_{si}$ and $R_{li}$ must be left
blank.

### 2.4.3 Model outputs

In Table 3, the main outputs of SCOPE are listed. The general output of SCOPE includes 1) spectral simulations of radiance in the viewing direction and upward flux for the whole upper hemisphere from optical to thermal domain including fluorescence 2) radiation budget, such as incoming and outgoing radiation for shortwave from 0.5 to 2.5 $\mu$m and longwave from 2.5 to 50
$\mu$m 3) fluxes such as sensible heat, latent heat and the ground heat flux for canopy, soil and the combined system and 4) canopy absorption, such as absorbed PAR by chlorophyll. Most of the stored outputs of SCOPE are for the whole canopy, although similar variables of leaves are also computed internally in SCOPE.





**Table 3.** SCOPE outputs

| output | description | unit |
|---|---|---|
| **spectral simulation** | | |
| Eout_spectum | hemispherical leaving irradiance | $[\mathrm{Wm}^{-2}\mu\mathrm{m}^{-1}]$ |
| Lo_spectrum | radiance in the viewing direction | $[\mathrm{Wm}^{-2}\mu\mathrm{m}^{-1}\mathrm{sr}^{-1}]$ |
| fluorescence | fluorescence radiance in the viewing direction | $[\mathrm{Wm}^{-2}\mu\mathrm{m}^{-1}\mathrm{sr}^{-1}]$ |
| fluorescence_hemis | hemispheric leaving fluorescence irradiance | $[\mathrm{Wm}^{-2}\mu\mathrm{m}^{-1}]$ |
| reflectance | TOC reflectance in the viewing direction | [] |
| **vegetation** | | |
| aPAR | PAR absorbed by the vegetation | $[\mu\mathrm{mol}\ \mathrm{m}^{-2}\mathrm{s}^{-1}]$ |
| aPARbyCab | PAR absorbed by chlorophyll | $[\mu\mathrm{mol}\ \mathrm{m}^{-2}\mathrm{s}^{-1}]$ |
| aPARbyCab_en | PAR energy absorbed by chlorophyll | $[\mathrm{Wm}^{-2}]$ |
| Photosynthesis | canopy photosynthesis rate | $[\mu\mathrm{mol}\ \mathrm{m}^{-2}\mathrm{s}^{-1}]$ |
| LST | black-body radiometric land surface temperature | [K] |
| **fluxes** | | |
| Rnctot | Net radiation of canopy | $[\mathrm{Wm}^{-2}]$ |
| lEctot | Latent heat flux of canopy | $[\mathrm{Wm}^{-2}]$ |
| Hctot | Sensible heat flux of canopy | $[\mathrm{Wm}^{-2}]$ |
| Actot | Net photosynthesis of canopy | $[\mathrm{Wm}^{-2}]$ |
| Tcave | Average canopy temperature | $[^{\circ}\mathrm{C}]$ |
| Rnstot | Net radiation of soil | $[\mathrm{Wm}^{-2}]$ |
| lEstot | Latent heat flux of soil | $[\mathrm{Wm}^{-2}]$ |
| Hstot | Sensible heat flux of soil | $[\mathrm{Wm}^{-2}]$ |
| Gtot | Soil heat flux | $[\mathrm{Wm}^{-2}]$ |
| Tsave | Average soil temperature | $[^{\circ}\mathrm{C}]$ |
| Rntot | Total net radiation | $[\mathrm{Wm}^{-2}]$ |
| lEtot | Total latent heat flux | $[\mathrm{Wm}^{-2}]$ |
| Htot | Total sensible heat flux | $[\mathrm{Wm}^{-2}]$ |
| **radiation** | | |
| ShortIn | Incoming shortwave radiation | $[\mathrm{Wm}^{-2}]$ |
| LongIn | Incoming longwave radiation | $[\mathrm{Wm}^{-2}]$ |
| HemisOutShort | hemispherical outgoing shortwave radiation | $[\mathrm{Wm}^{-2}]$ |
| HemisOutLong | hemispherical outgoing longwave radiation | $[\mathrm{Wm}^{-2}]$ |
| Lo | radiance in observation direction | $[\mathrm{Wm}^{-2}\mathrm{sr}^{-1}]$ |
| Lot | thermal radiance in observation direction | $[\mathrm{Wm}^{-2}\mathrm{sr}^{-1}]$ |
| Lote | emitted radiance in observation direction | $[\mathrm{Wm}^{-2}\mathrm{sr}^{-1}]$ |





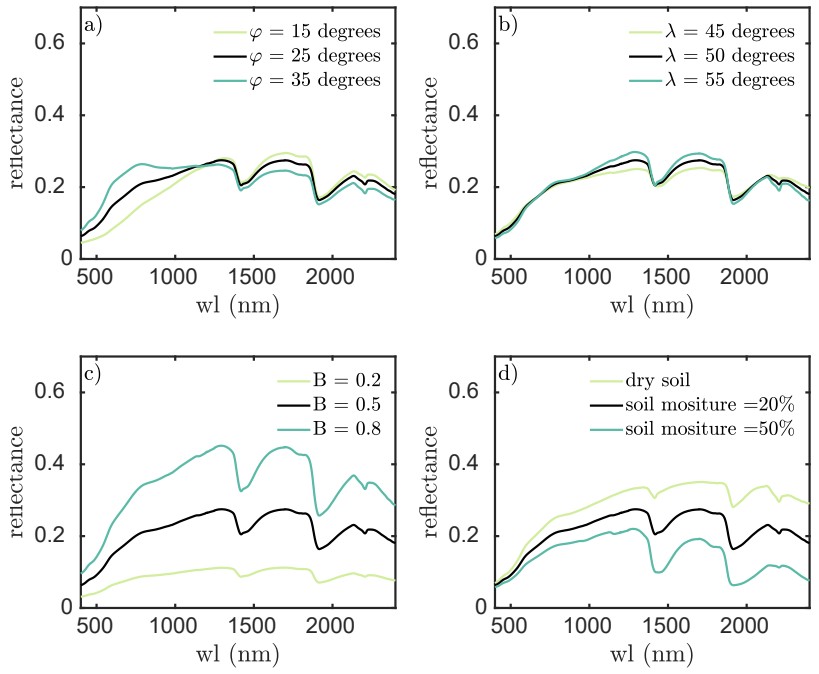

**Figure 2.** Reflectance simulations with the BSM model. The black curves in each panels are the same simulation.

# 3 Major improvements of SCOPE 2.0 compared with SCOPE

## 3.1 Implementation of the BSM soil reflectance model

In the first published version of SCOPE, the soil reflectance spectrum was an input variable. The users should either provide a measured soil spectrum or select one from the soil reflectance library incorporated in the SCOPE model. In SCOPE 2.0, we provide the users the option to simulate soil reflectance thanks to the implementation of a soil reflectance model.

The Brightness-Shape-Moisture (BSM) model simulates the isotropic soil reflectance. This model is based on an empirical reflectance model of dry soil (Verhoef et al., 2018; Jiang and Fang, 2019) and incorporates the effects of soil moisture by using
the water film coating approach (Ångström, 1925; Yang et al., 2020b). To simulate reflectance of dry soil, the model requires soil brightness ($B$) and two spectral-shape related parameters ($\varphi$ and $\lambda$) as inputs. Soil moisture is necessary for simulating wet soil reflectance.

Fig. 2 shows the effects of the four parameters on soil reflectance. It is evident that soil brightness only affects the 'intensity' of soil reflectance and the 'shape' of soil reflectance is controlled by $\varphi$ and $\lambda$. Soil moisture affects reflectance intensity over
all wavelengths, but reflectance at the water absorption bands are more sensitive to soil moisture. Soil moisture effects on reflectance are considerably similar to the effects of soil brightness, and the soil is dark when it is wet, as explained in Lekner and Dorf (1988).





## 3.2 Inclusion of dynamic reflectance induced by the xanthophyll cycle

A new feature in SCOPE 2.0 is modelling the photochemical reflectance dynamics induced by the xanthophyll cycle at both

leaf and canopy levels. In the original leaf RTM Fluspect (Vilfan et al., 2016), leaf optical properties are determined by leaf biophysical properties. However, in natural conditions, the xanthophyll cycle that is involved in photo-protection mechanisms under excess light can provoke a change in reflectance and transmittance as the composition of the pigment pool is regulated. Changes in the de-epoxidation state (DEPS) of xanthophyll cycle pigments (e.g., violaxanthin and zeaxanthin) can be observed as changes in the leaf absorption of light with wavelengths between 500 to 570 nm. These spectral changes can be a good

remote sensing indicator of the photosynthetic efficiency. The photochemical reflectance index (PRI, $\frac{R_{570}-R_{531}}{R_{570}+R_{531}}$) proposed by Gamon et al. (1992) is a example of a measure for the effects of xanthophyll cycle pigments on the reflectance. It takes changes in reflectance at 531 nm to estimate DEPS with reflectance at 570 nm as a reference to correct changes in reflectance induced by other factors, such as sun-observer geometry.

Vilfan et al. (2018) incorporated the effects of the xanthophyll cycle on leaf optical properties in Fluspect and developed the

Fluspect-CX model. The main idea of Fluspect-CX is to use *in vivo* specific absorption coefficients for two extreme states of carotenoids, representing the two extremes of the xanthophyll de-epoxidation. A 'photochemical reflectance parameter' (Cx) is employed to describe the intermediate states as a linear mixture of these two states. Cx controls the specific absorption coefficient of carotenoids in a leaf, and thus affects leaf reflectance and transmittance.

The propagation of changes in leaf reflectance and transmittance induced by the xanthophyll cycle to TOC reflectance

is carried out with RTMz, which is largely similar to RTMf in the sense that both the xanthophyll cycle and fluorescence emission lead to small changes in (apparent) reflectance but for different spectral regions (i.e., 500-570 nm and 640-850 nm, respectively). RTMf and RTMz take fluorescence emission efficiency and Cx (simulated from the leaf biochemical model), respectively, as inputs, of which the magnitudes vary among individual leaves due to their ambient light intensities, temperature, etc. Fig. 3 depicts an example of the effects of Cx on the leaf and canopy reflectance as simulated by SCOPE 2.0 with the default

model inputs (Table 2). Although the effects on canopy reflectance seem small, it could be helpful to monitor the variation in DEPS.

Fig. 4 compares simulations of PRI in a day with SCOPE and SCOPE 2.0. In these illustrative simulations, the default model inputs are used except for the incoming radiation and solar zenith angles. The values of incoming radiation and solar zenith angles ($\theta_s$) are assigned according to the field measurements presented in Yang et al. (2020a) (i.e., on day 232 of the dataset in

the referred paper). The comparison demonstrates that the inclusion of dynamic reflectance induced by the xanthophyll cycle has a clear impact on the simulation of diurnal changes in PRI. In SCOPE, diurnal variation of PRI is mainly regulated by sun-observer geometry, since leaf biophysical properties and canopy structure are kept unchanged in a day. Because the BRDF effects on reflectance at 531 nm and 570 nm are similar, they cancel out in PRI, and the diurnal variation of PRI simulated with SCOPE is small. Compared with SCOPE, SCOPE 2.0 considers the changes in leaf pigment pool induced by the xanthophyll

cycle in response to the variation of incoming radiation besides the BRDF effects. The excessive incoming radiation in the



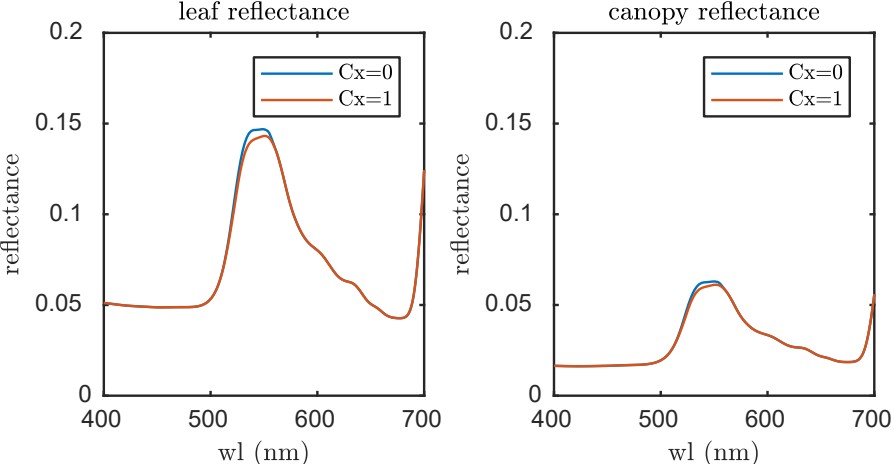

**Figure 3.** The effects of the xanthophyll cycle on leaf and canopy visible reflectance simulated with SCOPE 2.0 using the default model inputs (Table 2). Cx is a factor for the de-epoxidation state (DEPS) of xanthophyll cycle pigments.

midday leads to larger Cx values than in the morning and afternoon, and higher afternoon than morning temperatures to higher afternoon Cx, and thus more significant diurnal variation of PRI.

### 3.3 Adaption of the RTMs for multi-layer canopies

The original SCOPE model assumes that vegetation canopies are vertically homogeneous and horizontally infinite, as its radia-
tive transfer routines are based on the classical 1-D SAIL model (Verhoef, 1984). The vertical heterogeneity of leaf biophysical
and biochemical properties may have a large effect on the bi-directional reflectance, fluorescence and photosynthesis of veg-
etation canopies. To allow simulations of vertical heterogeneous canopies, Yang et al. (2017) modified the RTMs in SCOPE
and developed a new branch of SCOPE, called mSCOPE. SCOPE 2.0 incorporates the essence of mSCOPE on radiative trans-
fer modelling and adapts the capability to simulate reflectance, fluorescence and photosynthesis of vertically heterogeneous
canopies.

RTMs in SCOPE 2.0 remain structurally the same with the original SCOPE. However, a more general solution of the radiative
transfer problems is used. Compared to the classic SAIL analytical solution, SCOPE 2.0 (and mSCOPE) employs the adding
method to solve the radiative transfer problems. The application of the adding method for TOC reflectance simulation is given
in Verhoef (1985). Yang et al. (2017) extended this method to calculate the radiative flux profiles in the canopy. The procedure
is summarized as follows: 1) divide the vertical layer into $n$ thin homogeneous layers; 2) start from the bottom homogeneous
layer, calculate the surface reflectance of the combined system of the bottom surface (e.g., soil) and this layer; 3) add a new
homogeneous vegetation layer above the surface of the previous system in step 2, and calculate the surface reflectance of the
new system; 4) repeat step 3 until all homogeneous layers are added; 5) Once the surface reflectance at each vertical level is
obtained, the fluxes profile can be computed from top to bottom, given the incident fluxes at top of the canopy. For the radiative





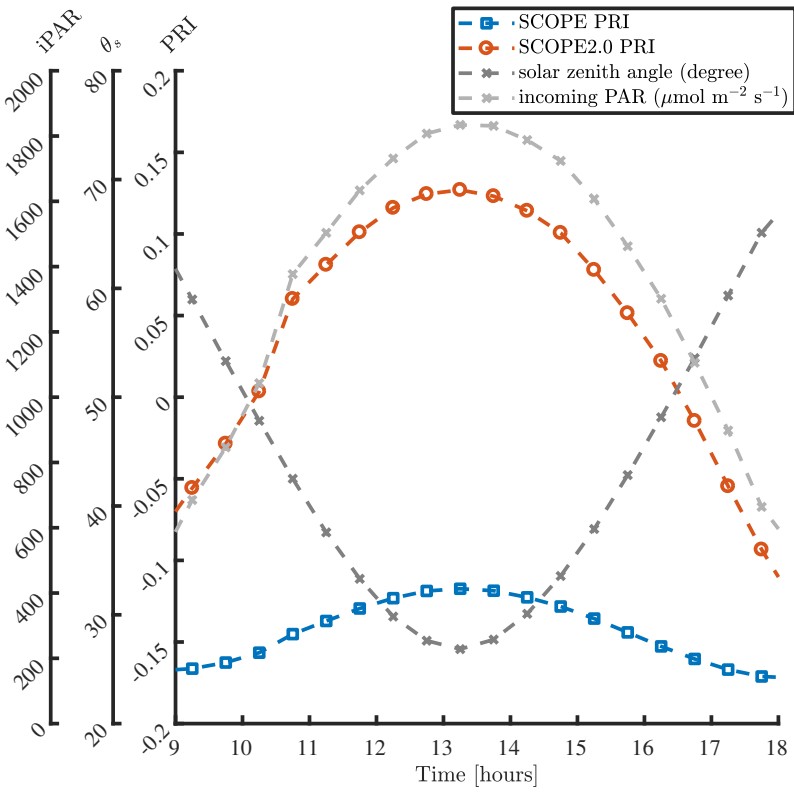

**Figure 4.** The effects of the xanthophyll cycle on diurnal PRI variation by comparing simulations from SCOPE and SCOPE 2.0. The default model inputs are used except for the incoming radiation and solar zenith angles, which are values in a representative sunny day (two grey lines). Note that SCOPE does not include the effects of the xanthophyll cycle on TOC reflectance while SCOPE 2.0 does.

transfer of fluorescence and thermal radiation, the emission from leaves and soil should be included as extra radiation sources besides the incident fluxes at top of the canopy. In SCOPE 2.0, the value of $n$ is set as 10 times LAI rather than a fixed value of 60 in mSCOPE, because this ensures the LAI of one elementary layer is small enough (i.e., iLAI$< 0.1$), and the use of less elementary layers improves the computational efficiencies of the RTMs.

### 3.4    An alternative way to estimate the ground heat flux

In SCOPE the ground heat flux is calculated for the sunlit and shaded soil. In the original SCOPE model, this was either a constant fraction of 0.35 of the net radiation on the soil, or calculated with the force restore method of Bhumralkar (1975). SCOPE 2.0 offers an alternative way to estimate the ground heat flux as a function of the soil temperature time series with the method of Wang and Bras (1999). The ground heat flux is determined by the gradient of soil temperature in the profile underneath the soil surface. The subsurface is outside the model domain of SCOPE, and therefore the soil temperature gradient



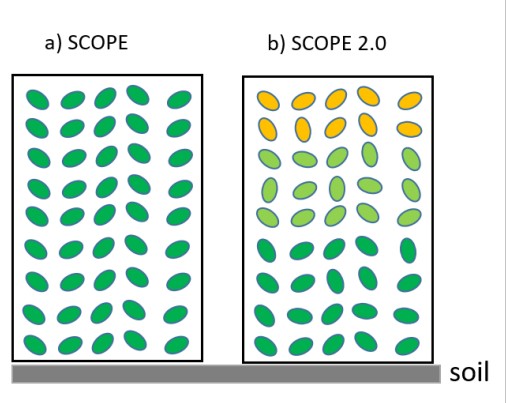

**Figure 5.** Representations of canopies in SCOPE and in SCOPE 2.0. In SCOPE, a canopy consists of leaves with identical biochemical and biophysical properties, and the leaf angle distribution is vertically invariant, while in SCOPE 2.0, both leaf properties and leaf angles can vary vertically.

is not simulated. However, this vertical gradient may equivalently be expressed the by the half-order time-derivative of the surface temperature (Wang and Bras, 1999). This enables the estimation of $G$ from the time-history of the surface temperature:

$$G(t) = \Gamma/\sqrt{\pi} \int_{t0}^{t} \frac{T(s)}{t-s} ds \tag{3}$$

where $T$ is the soil temperature at time $s$, $\Gamma$ [J m$^{-2}$ s$^{-1/2}$ K$^{-1}$] is the thermal inertia of the soil, calculated from physical properties of the soil:

$$\Gamma = \sqrt{c_s \cdot \rho_s \cdot \lambda_s} \tag{4}$$

where $c_s$ is the volumetric heat capacity of the soil [J kg$^{-1}$K$^{-1}$], $\rho_s$ the soil bulk density [kg m$^{-3}$], and $\lambda_s$ [J m$^{-1}$ s$^{-1}$ K$^{-1}$] the heat conductivity of the soil. In SCOPE 2.0 a solution derived for a discrete time series of temperatures by Bennett et al. (2008) (Eq. A3 therein) was adopted:

$$G(t) = 2\Gamma/\sqrt{\pi} \sum_{i=1}^{12} \frac{T_{i+1} - T_i}{s_{i+1} - s_i} (\sqrt{t-s_i} - \sqrt{t-s_{i+1}}) \tag{5}$$

This approach is only meaningful if consecutive simulations are carried out in a time series, in which the diurnal variation of temperature is reproduced (at least one simulation per three-hour time step). The approximation of $G = 0.35 R_{ns}$ should be used for cases in which the state of the soil heat reservoir cannot be known, for example, if simulations are carried out for pixels in a satellite image taken at a single moment in time.




### 3.5 Improvements in energy balance closure

The energy balance loop starts by simulating the radiative transfer of internally generated radiation with initial estimates of component temperatures, followed by the calculation of aerodynamic and stomatal resistances (and photosynthesis), and the fluxes $H$, $\lambda E$ and $G$. Finally, new estimates of the component temperatures are calculated from the value of the energy balance closure error ($\Delta E$) per leaf and soil element. Newton's method is used to estimate the new temperatures, which are the starting point for the next iteration in the loop.

$$T_{new} = T_{old} + W \cdot \frac{e_{bal}}{\delta e_{bal}/\delta T} \tag{6}$$

where $\delta e_{bal}/\delta T$ is the first derivative of the energy balance closure error to temperature, and $W$ is a weighting for the step size. The derivative is estimated analytically:

$$T_{new} = T_{old} + W \frac{e_{bal}}{\rho \cdot c_p/r_a + \rho \cdot \lambda \cdot M_{H2O}/M_{air}/p \cdot s/(r_a + r_s) + 4\varepsilon\sigma(T_{old} + 273.15)^3} \tag{7}$$

The derivative is estimated independently for all leaf and soil elements. In the estimate, it is assumed that the incident irradiance on the leaves (or soil) does not change. This is an approximation. The internally (in the canopy) generated incident irradiance depends on the temperature of the neighbouring leaves, which is updated in the next iteration step as well. Further, it is assumed that the resistances $r_a$ and $r_s$ do not change between iteration steps. This is an approximation as well, as both depend on leaf and soil temperature. Although these interactions cannot be resolved analytically, Eq. 7 is a sufficiently accurate approximation of the first derivative to obtain rapid energy balance convergence. Iterations continue until the maximum absolute closure error of all leaf and soil elements is less than 1 Wm$^{-2}$, and this is usually achieved in less than 10 iteration steps. If energy balance closure is not achieved after 10 steps, then the weighting coefficient $W$ is gradually decreased from 1 (i.e. smaller update steps) to avoid the updated temperatures bouncing around the solution.

In earlier versions of SCOPE, a similar equation to Equation 7 has been used to update temperature in the energy balance loop. However, the partial derivative of latent heat flux to temperature was not included in the equation. The improvement in SCOPE 2.0 has substantially reduced the number of required iterations due to a more complete estimate of the derivative.

### 3.6 Angular aggregation of sunlit leaves

In the energy balance routine, the number of sunlit leaf elements that are considered is 13 leaf zenith times 36 leaf azimuth times $10 \times$ LAI layers, while the number of shaded leaf elements is $10 \times$ LAI since leaf angles play no role for the interactions between a leaf and radiative fluxes when a leaf is illuminated by isotropic diffuse light. Solving the energy budget for all these elements separately means that closure of energy balance should be achieved for each element, and this is computationally demanding. SCOPE 2.0 offers the possibility to simulate the non-radiative energy fluxes, photosynthesis and gas exchange for all inclination and azimuth angles of the sunlit leaves combined (the 'lite' option). This involves an aggregation (weighted averaging) of net radiation over all leaf angles, before entering the energy balance loop. One effective leaf for the $13 \times 36$ sunlit leaf classes is used for each layer. The resulting number of elements is $10 \times$ LAI for the sunlit leaves, and $10 \times$ LAI for the shaded leaves. This significantly reduces the computation time of the energy balance routine.





The consequence of this internal aggregation is that the all sunlit leaves in a layer will have an identical temperature, gas exchange, photosynthesis rate, chlorophyll fluorescence emission efficiency, and latent and sensible heat flux, independent of their inclination towards the sun. Figs. 6 and 7 present examples for the effects of the angular aggregation on the profiles of leaf temperature and photosynthesis simulations, respectively. In these simulations, the default model inputs are used (Table 2).

Due to the simplifications in the energy balance and biochemical part in the lite mode, the layer average temperatures become slightly higher for both sunlit and shaded leaves (Fig. 6). A slight difference in photosynthetic production between the lite on and lite off modes can be found for sunlit leaves, but the difference for shaded leaves is negligible (Fig. 6). The photosynthetic production simulation for the whole canopy changes by about 0.7 $\mu$mol m$^{-2}$s$^{-1}$ (4%) when the lite mode is activated. The differences in leaf temperature and photosynthesis are apparently affected by the incoming radiation, leaf biochemistry, canopy

structure and other model inputs. The implementation of the lite mode might be helpful for estimation of the added value of consideration of various leaf orientations in a canopy in comparison of the simpler one-big-leaf or two-big-leaf models (Dai et al., 2004; Luo et al., 2018).

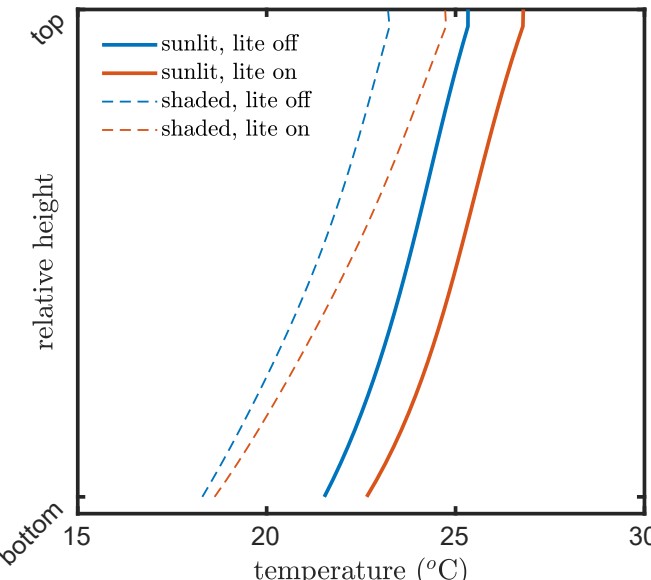

**Figure 6.** Layer average kinetic temperatures of the leaves in the vegetation canopy simulated by SCOPE 2.0 with the 'lite' representation on (red) and off (blue) of the vegetation, for sunlit (solid lines) and shaded (dash lines) leaves.

With the lite option switched on, the emitted thermal and fluorescence radiation is calculated for layer-average temperature and emission efficiency, respectively, albeit separately for the sunlit and shaded portions. The aggregated layer properties will

propagate into the simulation of fluorescence and surface brightness temperature ($T_b$) as observed above the canopy. Fig. 8 presents an example for the effects of the angular aggregation on fluorescence and $T_b$ simulation with the default model inputs. It is worth noting that the RTMs are all carried out with the original representation of the canopy, thus with $13 \times 36$ leaf orientations per layer. This means that the lite mode has no influence on reflectance, net radiation in the optical domain, and





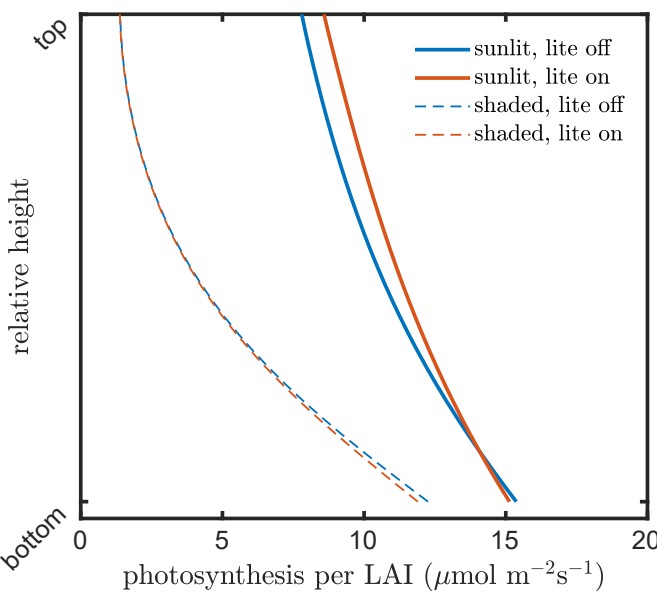

**Figure 7.** Layer photosynthesis per LAI in the vegetation canopy simulated by SCOPE 2.0 with the 'lite' representation on (red) and off (blue) of the vegetation, for sunlit (solid lines) and shaded (dash lines) leaves.

absorbed photosynthetically active radiation (APAR) by leaves. Moreover, the directionality and hotspot is still simulated (Fig.

370  8).

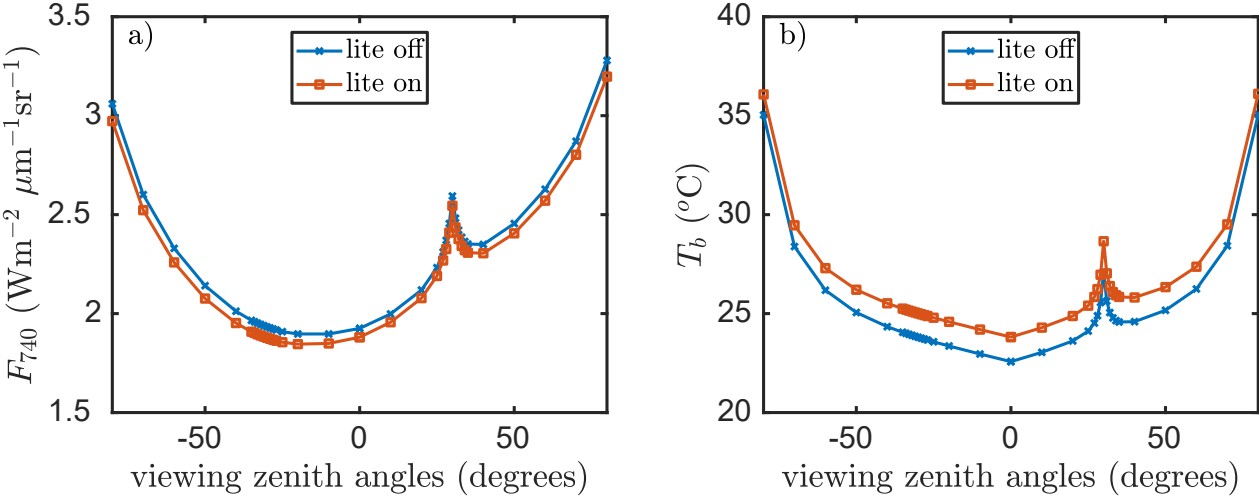

**Figure 8.** (a) Top-of-canopy fluorescence at 740 nm ($F_{740}$) and (b) surface brightness temperature ($T_b$) simulated by SCOPE 2.0 with the 'lite' representation on (red) and off (blue) of the vegetation versus viewing angles in the principle plane.





## 3.7 Improvements in the computational efficiency

In SCOPE 2.0, substantial reductions in computation have been achieved compared to SCOPE 1.70 (Table 4). In a test case of 100 scenarios run by SCOPE 2.0 using a regular PC, the computation time is 12.8% (lite option off) or 7% (lite option on) of the same 100 scenarios run by SCOPE 1.70. The reduction of computation time is due to (in order of decreasing

contribution) (1) a more efficient energy balance closure, (2) more efficient saving of output (initially as binary files, later converted to CSV), (3) the overall reduction of the number of layers (from 60 to 10×LAI) and (4) introducing the mSCOPE radiative transfer equations, which allows for better re-use of earlier calculated quantities. A further reduction in computation time can be achieved when switching off the temperature correction of biochemical parameters (such as $V_{cmo}$) with the option 'tempcor', due to a more rapid convergence of the energy balance loop (not shown).

**Table 4.** Breakdown of computation time to the most computationally intensive modules in SCOPE 1.70 and SCOPE 2.0 expressed as seconds per 100 simulations.

| module | computation time (s/100 simulation) | | |
|---|---|---|---|
| | SCOPE1.70 | SCOPE2.0 | |
| | | lite off | lite on |
| SCOPE self time | 0.87 | 0.51 | 0.51 |
| BSM | / | 0.28 | 0.28 |
| Fluspect | 3.5 | 1.29 | 1.26 |
| RTMo | 5.75 | 2.53 | 2.53 |
| RTMt_planck | 33.89 | 3.69 | 0.91 |
| RTMf | 14.1 | 0.58 | 0.63 |
| RTMz | / | 0.62 | 0.64 |
| importdata | 0.6 | 0.03 | 0.03 |
| ebal | 78.8 | 7.06 | 1.26 |
| output | 32.87 | 2.3 | 2.3 |
| the rest | 7.22 | 3.82 | 3.5 |
| **total** | **177.6** | **22.7** | **13.85** |

## 380 3.8 Additional outputs

In addition to the output of the original SCOPE model, more model output parameters are produced and stored in SCOPE 2.0, considering users' needs. In Table 5, the outputs available in SCOPE 2.0 but not in the original SCOPE model are presented. Nevertheless, it is worth noting that all the outputs produced in SCOPE 2.0 can also be computed in the original SCOPE with little effort, although they are not stored as outputs.





In the original SCOPE model, TOC reflectance spectral simulation in the viewing direction is provided as an output. It is computed as

$$R = \frac{\pi L_o}{E_{sun} + E_{sky}} \tag{8}$$

where $L_o$ is the radiance in the viewing direction excluding fluorescence contribution, and $E_{sun}$ and $E_{sky}$ the incoming direct solar and diffuse sky irradiance.

In practice, many users do not have measurements of $E_{sun}$ and $E_{sky}$ or atmospheric properties available for inputs but the fraction of diffuse light ($f_{sky}$). Therefore, we provide the directional reflectance factors of the surface as outputs: $r_{so}$, $r_{sd}$, $r_{dd}$ and $r_{do}$. The two-letter subscripts indicate the incident and outgoing fluxes types: $d$ referring to the diffuse fluxes, $s$ referring to the direct solar flux and $o$ referring to the flux in the viewing direction. These four reflectance factors are independent of the incoming irradiance but are optical properties of the soil-vegetation system. The canopy reflectance in the viewing direction

can be estimated as

$$R = (1 - f_{sky})r_{so} + f_{sky}r_{do} \tag{9}$$

Furthermore, the radiance in the viewing direction including the fluorescence contribution is provided, which allows computing the apparent reflectance of a vegetation canopy besides the true reflectance.

We include several fluorescence variables as outputs to help to better interpret fluorescence signals in SCOPE 2.0, besides
fluorescence at top of canopy. Because fluorescence produced by all photosystems is considered to have a more direct relationship with canopy GPP (Yang and Van der Tol, 2018; Van der Tol et al., 2019a), we include it in the outputs. This allows us to compute an important variable: the fluorescence scattering coefficient, which is defined as:

$$\sigma_F = \pi L_o^F / E_F; \tag{10}$$

where $E_F$ is the total emitted fluorescence irradiance by all photosystems, calculated as the canopy integration of the product
of absorbed photosynthetically active radiation by chlorophyll, the fluorescence yield, and the (constant) spectral shape of chlorophyll fluorescence. The coefficient $\sigma_F$ is sometimes referred to as the 'escape probability' in the literature. It can be used to correct the fluorescence for both sun-observation geometry and reabsorption of fluorescence in the canopy in order to estimate a canopy-effective fluorescence yield (Yang et al., 2020a).

The biochemical model quantifies the energy partitioning into different pathways and computes their light use efficiencies
at leaf scale. The energy partitioning concept is applied to the whole canopy. By taking the weighted average values of the efficiencies of individual leaves, we obtain canopy electron transport rate and non-photochemical quenching (NPQ), which describes the effective photosynthetic light use efficiency and the effective efficiency of the heat dissipation pathway of the canopy (Maxwell and Johnson, 2000). These variables are direct indicators of the physiological status of the whole canopy.





**Table 5.** Additional outputs in SCOPE 2.0

| output | description | unit |
|---|---|---|
| **spectral simulation** | | |
| Lo_spectrum_inclF | radiance in the viewing direction including fluorescence | $[\mathrm{Wm}^{-2}\mu\mathrm{m}^{-1}\mathrm{sr}^{-1}]$ |
| rso, rsd, rdd, rdo | four canopy reflectance factors | [] |
| sigmaF | fluorescence scattering coefficient | [] |
| **fluorescence_scalars** | | |
| LFtot | spectrally integrated observed fluorescence | $[\mathrm{Wm}^{-2}\mathrm{sr}^{-1}]$ |
| EFtot | spectrally and hemispherically integrated fluorescence | $[\mathrm{Wm}^{-2}]$ |
| EFtot_RC | spectrally and hemispherically integrated fluorescence corrected for reabsorption | $[\mathrm{Wm}^{-2}]$ |
| **vegetation** | | |
| Electron_transport | canopy electron transport rate | $[\mu\mathrm{mol}\ \mathrm{m}^{-2}\mathrm{s}^{-1}]$ |
| NPQ_energy | energy dissipated as non-photochemical quenching | $[\mathrm{Wm}^{-2}]$ |

## 4 Conclusions

We presented a significantly improved version of the Soil-Canopy Observation of Photosynthesis and Energy fluxes (SCOPE) model. SCOPE 2.0 simulates the energy balance fluxes of net radiation, sensible and latent heat flux, ground heat flux and photosynthesis, as well as hyperspectral radiance in the optical and thermal domain including the contribution of fluorescence.

The improved computational efficiency and model stability make the model a suitable tool for routine estimation of fluxes and satellite signals in homogeneous vegetation canopies with an understory and overstory, or multi-layer structure. The new 420 features also include the simulation of a subtle change in the reflectance due to the xanthophyll cycle dynamics in the range of 500-600 nm, allowing better investigating vegetation physiology under various weather conditions.

*Author contributions.* CT and PY designed the model general structure of SCOPE2.0. WV and PY developed the BSM model and the multi-layer vegetation parameterization. CT, PY and EP wrote the model code and performed the simulations. EP and CT wrote the online model documentation. PY and CT prepared the manuscript with contributions from all co-authors.

*Code availability.* The source code is freely available to users on Github (https://github.com/Christiaanvandertol/SCOPE2). SCOPE is written in Matlab and is compatible with versions of 2013a and later. A compiled version is available for the Matlab Runtime Compiler 2019a.

*Competing interests.* The authors declare that there is no competing interests.





*Acknowledgements.* CT was funded by ESA under Contract No.4000122680/17/NL/MP, CNN2. PY was funded by the Netherlands Organization for Scientific Research, grant ALWGO.2017.018. EP has received funding from the European Union's Horizon 2020 research and
innovation programme under the Marie Sklodowska-Curie grant agreement No 721995. Many users contributed with their feedback and suggestions. Particular thanks to: Ari Kornfeld, Christian Frankenberg, Joe Berry, Albert Olioso, Jerome Démarty, Federico Magnani, Jose Moreno, Yves Goulas and Marco Celesti.



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
