# Peer review of "SCOPE 2.0: A model to simulate vegetated land surface fluxes and satellite signals"

_Geoscientific Model Development, 2020_

## Short Comment (SC1) · 14 Nov 2020

Dear authors,

in my role as Executive editor of GMD, I would like to bring to your attention our Editorial version 1.2:

https://www.geosci-model-dev.net/12/2215/2019/

This highlights some requirements of papers published in GMD, which is also available on the GMD website in the 'Manuscript Types' section: http://www.geoscientific-model-development.net/submission/manuscript_types.html

In particular, please note that for your paper, the following requirement has not been

met in the Discussions paper:

- Code must be published on a persistent public archive with a unique identifier for the exact model version described in the paper or uploaded to the supplement, unless this is impossible for reasons beyond the control of authors. All papers must include a section, at the end of the paper, entitled "Code availability". Here, either instructions for obtaining the code, or the reasons why the code is not available should be clearly stated. It is preferred for the code to be uploaded as a supplement or to be made available at a data repository with an associated DOI (digital object identifier) for the exact model version described in the paper. Alternatively, for established models, there may be an existing means of accessing the code through a particular system. In this case, there must exist a means of permanently accessing the precise model version described in the paper. In some cases, authors may prefer to put models on their own website, or to act as a point of contact for obtaining the code. Given the impermanence of websites and email addresses, this is not encouraged, and authors should consider improving the availability with a more permanent arrangement. Making code available through personal websites or via email contact to the authors is not sufficient. After the paper is accepted the model archive should be updated to include a link to the GMD paper.

As GitHub is not a persistent archive, please provide for the presented source code a persistent identifier. As explained in https://www.geoscientific-model-development.net/about/manuscript_types.html the preferred reference to this release is through the use of a DOI which then can be cited in the paper. For projects in GitHub a DOI for a released code version can easily be created using Zenodo, see https://guides.github.com/activities/citable-code/ for details.
Finally note, that according to our new Editorial (v1.2) all data and analysis / plotting scripts should be made available.

Yours, Astrid Kerkweg

---

## Short Comment (SC2) · 7 Dec 2020

Dear executive Editor, Thank you for this feedback. We have published the model code in a persistent public archive, with the following DOI: https://doi.org/10.5281/zenodo.4309327 We will also make the code to reproduce the figures available as supplementary information. On behalf of all authors, Christiaan van der Tol

---

## Referee Comment (RC1) · Yelu Zeng (Referee) · 15 Mar 2021

Thanks for inviting me to review this manuscript. Earlier versions of SCOPE have been widely used in SIF, GPP, energy balance and thermal signal simulations. This paper mainly summarized the recent progress of the revised version, SCOPE 2.0, which includes (1) the consideration of multi-layer vertical variation of chlorophyll content and leaf angle distributions; (2) the adoption of the BSM soil reflectance model to account for the soil moisture; (3) the impact of the xanthophyll cycle on the leaf-canopy reflectance and (4) speed acceleration optimization. I believe these advances are of interest to the remote sensing and ecology modeling communities, and thus this paper matches the scope of the GMD journal very well. Overall, this paper is well written

and structured. I have a few comments from the perspective of a SCOPE user, and the authors may choose to consider or not according to the long-term plan of SCOPE improvements and the amount of effort needed.

1. P21, L390: Currently in SCOPE2.0, atmospheric properties are the input parameters that determine the proportion of the direct and diffuse solar radiation. As the authors mentioned in Line 390, for the simulation of a specific site (e.g., some Fluxnet and PhenoCam sites), PAR and diffuse PAR ratio are usually available instead of the atmospheric properties. This makes the simulation of SIF and photosynthesis to be difficult at the diurnal or seasonal cycle with different diffuse PAR ratio. It would be more convenient for users in ecology community, if PAR and diffuse PAR ratio could be used as input parameters in SIF and photosynthesis simulations.

2. Second is about the validation. As far as I know, there is rare literature about the validation of SCOPE over high productivity areas with GPP > 40 umol CO2 m-2 s-1. From my experience of using SCOPE to simulate GPP of soybean at the Corn Belt in the US and in summer, it is very difficult to be able to achieve the GPP simulations to be larger than 40 umol CO2 m-2 s-1 with the field PAR, temperature, chlorophyll content, etc measurements, unless we set the unmeasured Vcmax to be larger than 200 umol m-2 s-1. Of course, this is unreasonable. Validation of SCOPE simulated GPP over high productivity areas would give more confidence and guidance to the SCOPE users in ecology community.

3. P 15, Fig. 4: For the xanthophyll cycle, Vilfan et al (2018) only focused on the leaf scale. In fact, there are already many canopy-scale PRI field observations acquired during the plant stress in recent studies. Showing the capability (and good performance) of SCOPE2.0 with field data to capture the plant stress by quick response (and accurate simulation) of PRI (or CCI) would be more interesting and convincing than model comparisons if possible.

4. P17, L 345-350: Usually it is difficult to determine how many layers should be set

in SCOPE for a specific vegetation species, e.g., corn with narrow and long inclined leaves, or taro with big leaves but not many layers. In my understanding, the layers in SCOPE and the leaf layer in reality are different. For example, if there is only one flat big leaf over the ground, sensors can always observe the hot-spot effect in all viewing directions, while this kind of situations are difficult to consider in model settings. Of course, this is the gap between abstract models and nature in reality. The current model is correct under general assumptions of radiative transfer modelling, while caveats and more guidance may be still needed for users to correctly use the model and achieve accurate simulations.

5. P16, Fig. 5: Does SCOPE2.0 have the capability to simulate the scene with different leaf sizes at different layers? Even if the leaf size is the same for all layers, the hot spot factor or leaf specific dimension could vertically vary with different multi-layer leaf angle distributions (Kuusk 1991). Seems this issue was not considered and the hot-spot effect was not evaluated or discussed in mSCOPE and SCOPE2.0 manuscripts. Not quite sure how large are the uncertainties by this issue to the reflectance around the hot spot directions. If fixed as one value, I suppose the multi-layer hot spot factor could be closer to the hot spot factor of the upper layer instead of the vertically averaged value. Maybe uncertainties by this issue could be evaluated by 3D multi-layer simulations.

Kuusk, A. (1991). The hot spot effect in plant canopy reflectance. In Photon-Vegetation Interactions (pp. 139-159). Springer, Berlin, Heidelberg.

6. P12, L230: The BSM model which simulates the isotropic soil reflectance was adopted in this study. For sparse vegetation canopies such as shrubland with low fractional vegetation cover and considerable soil roughness, the soil anisotropy and hot-spot effects are also important to the canopy reflectance for the chlorophyll content and leaf area index retrievals. Hope in the future the soil anisotropic model, e.g., the Hapke model, could be incorporated in the SCOPE framework at least for the soil single scattering contribution.

7. A discussion paragraph or section maybe needed to show the future directions of SCOPE improvements. Recently the leaf specular reflection has been reported to considerably contribute to the canopy reflectance especially over needle leaf forest, while this effect seems was not well considered in the current version. Besides, the 3D complex forest structures which can cast crown-scale dark shadows may also be the challenge for the current 1D models.

8. Seems the canopy coverage Cv was considered in the code of SCOPE2.0, while was not mentioned in the current manuscript.

9. Congratulations to the authors for the several important advances of SCOPE2.0, and I can foresee that this paper as a milestone of SCOPE will have considerable impact on the remote sensing and SIF community. Not all my concerns need to be addressed this time according to the feasibility and available dataset, and some of them could be in the discussion. The accurate description and guidance of the model can better meet the users' needs and expectations. The well validation of the model by field observations can help to bridge the gap between abstract models (modeler community) and complex reality (user community with observations), and is also helpful for the future model improvements.

---

## Referee Comment (RC2) · Anonymous Referee #2 · 29 Mar 2021

This study improves the widely-used radiative transfer and biophysical model SCOPE by implementing 1) soil reflectance simulation, 2) xanthophyll cycle modulation, 3) vertical variations of vertical properties, 4) dynamic ground heat flux simulation, 5) a full energy balance closure solution, and 6) multiple strategies for computational efficiency. These improvements are significant advances and I believe the proposed SCOPE 2.0 will benefit the vegetation remote sensing community. The paper is well written and I only one major comment followed by several minor comments.

Major comment:

1. While the improved algorithms are well described, the performance/effects of the new algorithms are not fully demonstrated.

(1) Can you compare TOC reflectance, GPP and SIF between using a vertically variant Cab and using an invariant Cab? This is very interesting as I see many studies, including ESA's products, interpret canopy chlorophyll content as the product of LAI and Cab without considering the vertical variation of Cab. SCOPE 2.0 can help us understand this impact.

(2) Can you compare typical diurnal cycles of G between G = 0.35Rn and the new parameterization?

(3) Can you use figures/tables to show 1) how energy balance closure is improved by using the new iteration algorithm, and 2) why Eq. 7 is a sufficiently accurate approximation?

Minor comments:

1. L16: I would suggest add some introduction of other models that can simulate radiative transfer and fluxes and provide distinct feature of SCOPE comparing to these models.

2. L87: SCOPE lacks the consideration of clumping effect, right? If so, I suggest add some words about that so that users can keep it in mind.

3. Table 1: The term "each leaf" is unclear. How many "leaves" in SCOPE 2.0? 13*36*n for sunlit and shaded, respectively?

4. L128: What type of aerodynamic resistance scheme is used in SCOPE 2.0? Series or parallel?

5. L180: Why is z "typically 2.5 times the vegetation height"? If we use meteorological data from site data or reanalysis data, they are fixed, right?

6. Table 2: Is there a relationship between Cab and Cs because senescenced leaves have lower Cab? Is there a relationship between Vcmax and Cab in terms of vertical variation? Why is Ball-Berry intercept parameter missed? Are their emissvity parameters?

7. L218: I'm confused here. If we need to conduct a time series simulation or spatial simulation, do we need to provide variant tau and rho parameters?

8. L227: While canopy FPAR can be obtained from outputs by FPAR = APAR/PAR, how can we get FPAR for leaves (sunlit/shaded at different layers)?

9. Table 3: What's the relationship between LST, Tcave and Tsave? Is this LST term comparable to ground/satellite estimates?

10. Section 3.3: How to input multi-layer vegetation parameters seems not mentioned. Also curious if vertical variation of meteorological data is modeled?

11. Figure 5: This figure is not cited in the text.

12. Figure 8: Does the bias indicate that the lite option is not suitable for thermal remote sensing? I think such clarification might be useful to users.

13. L418. While the "improved computational efficiency" is shown in Table 4, the "improved model stability" does not have evidence in the manuscript.

14. L419. The topic "understory and overstory" is never mentioned in the manuscript. Does SCOPE 2.0 has understory and overstory LAI separated?

---

## Author Comment (AC2) · 23 Apr 2021

**SCOPE 2.0: A model to simulate vegetated land surface fluxes and satellite signals**

**Response to reviewers' comments (Referee #2)**

*Peiqi Yang, Egor Prikaziuk, Wout Verhoef, Christiaan van der Tol\**

*April, 2021*
* * *
**General comments**

This study improves the widely-used radiative transfer and biophysical model SCOPE by implementing 1) soil reflectance simulation, 2) xanthophyll cycle modulation, 3) vertical variations of vertical properties, 4) dynamic ground heat flux simulation, 5) a full energy balance closure solution, and 6) multiple strategies for computational efficiency. These improvements are significant advances and I believe the proposed SCOPE 2.0 will benefit the vegetation remote sensing community. The paper is well written and I only one major comment followed by several minor comments.

Dear reviewer, We thank you for your precious time as well as your constructive and positive comments. Please find ou

**Major comments**

1. While the improved algorithms are well described, the performance/effects of the new algorithms are not fully demonstrated.

(1) Can you compare TOC reflectance, GPP and SIF between using a vertically variant Cab and using an invariant Cab? This is very interesting as I see many studies, including ESA's products, interpret canopy chlorophyll content as the product of LAI and Cab without considering the vertical variation of Cab. SCOPE 2.0 can help us understand this impact.

Response: We agree totally with the reviewer on the importance of vertical variation of leaf biophysical properties. Although the specific experiment/comparison is not included in this manuscript, we have already done this in the other paper about the mSCOPE model, which is a branch of the SCOPE model that includes vertical heterogeneity. In that work, we show the effects of vertical profiles of Cab on TOC reflectance, GPP, SIF and light use efficiency. To avoid repetition, we decided not to include such comparison in this manuscript, but added sentences clearly stating the comparison is available in the mSCOPE paper.

Yang, P., Verhoef, W., & van der Tol, C. (2017). The mSCOPE model: A simple adaptation to the SCOPE model to describe reflectance, fluorescence and photosynthesis of vertically heterogeneous canopies. Remote sensing of environment, 201, 1-11.

(2) Can you compare typical diurnal cycles of G between G = 0.35Rn and the new parameterization?

Response: With 35% parametrization the soil heat flux (Gtot) does not go below 0 W m-2, resulting in overestimation of the total annual sums.

[Figure]

(3) Can you use figures/tables to show 1) how energy balance closure is improved by using the new iteration algorithm, and 2) why Eq. 7 is a sufficiently accurate approximation?

Response: The older version 1.74 requires on average 59 iterations to close the energy balance, the 2.0 version requires on average 9. Furthermore, the standard deviations are dramatically different 59+/- 92 vs 9 +/- 9 iterations. This is largely due to the use the Eq. 7.

[Figure]

Eq. 7 is a linearization of the relation between temperature and energy balance error (i.e., please see the derivations followed by this response). This linearization is estimated analytically, which is much faster than calculating the derivative numerically. It is sufficient because it provides the slope, and thus a good update step of the temperature. The update is not exact, because the net radiation of the leaves also depends on the temperature of the other leaves (through the radiative transfer model). This cannot be solved analytically. The estimate is still sufficiently accurate, because it results in quick energy balance closure. The number of iteration steps in SCOPE 2.0 is significantly lower than in earlier versions of the model.

$$\delta e_{bal} = Rn - \lambda E - H \qquad (1)$$

$$\frac{\delta Rn}{\delta T} = \frac{\delta Rn_{sw}}{\delta T} + \frac{\delta Rn_{lw}}{\delta T} = 0 - \frac{\delta \sigma_{SB} * (T + 273.15)^4}{\delta T} = -4 * \sigma_{SB} * (T + 273.15)^3 \qquad (2)$$

$$\frac{\delta H}{\delta T} = \frac{\delta \rho * c_p. / r_a * (T_c - T_a)}{\delta T} = \rho * c_p. / r_a \qquad (3)$$

$$\frac{\delta \lambda E}{\delta T} = \frac{\delta \rho / (r_a + r_s) * \lambda * (q_i - q_a)}{\delta T} = \rho / (r_a + r_s) * \lambda * \frac{\delta q_i}{\delta T} \qquad (4)$$

$$q_i = 6.107 * 10^{\frac{7.5*T}{237.3+T}} \qquad (5)$$

$$\frac{\delta q_i}{\delta T} = s = q_i * ln(10) * \frac{7.5 * 237.3}{(237.3 + T)^2} \qquad (6)$$

**Minor comments:**

1. L16: I would suggest add some introduction of other models that can simulate radiative transfer and fluxes and provide distinct feature of SCOPE comparing to these models.

Response: We have added an introduction to about some other models, such as ACACIA, CUPID, SiB.

2. L87: SCOPE lacks the consideration of clumping effect, right? If so, I suggest add some words about that so that users can keep it in mind.

Response: Yes, it is right, although we are working on including this effect. As suggested, we have added a sentence stating the lack of clumping effects in SCOPE2.0.

3. Table 1: The term "each leaf" is unclear. How many "leaves" in SCOPE 2.0? 13*36*n for sunlit and shaded, respectively?

Response: We see the confusion here. To make it clearer, we changed "each leaf" to "individual leaves".

Yes, there are 13*36*n types of leaves in the canopy regardless whether they are sunlit and shaded. The leaves are different from each other by their orientation and leaf biophysical properties. In total, 13*36 types of leaf orientations are defined in SCOPE 2.0, and the biophysical properties of the leaves in the n vegetation layers can vary among the layers.

4. L128: What type of aerodynamic resistance scheme is used in SCOPE 2.0? Series or parallel?

Response: The leaves and soil are independently parallel sources, so the model is a multi-source model. Each leaf and soil element has three resistances in series: stomatal/soil surface (rs), leaf/soil laminar boundary (rb), within vegetation (rwc). These are parallel, merging to one level just above the vegetation. The resistance above the vegetation is common for all leaves and soil. The resistance scheme is described in more detail in Van der Tol et al. (2009) and Wallance and Verhoef (1997).

leaf i  --->rs -> rb -> rwc  ->

                                              ra -> air

leaf n ---> rs -> rb -> rwc -->

5. L180: Why is z "typically 2.5 times the vegetation height"? If we use meteorological data from site data or reanalysis data, they are fixed, right?

Response: The resistance scheme assumes that at this height (2.5*hc), the logarithmic wind profiles starts. The height should be set to the height of the meteorological tower, and this height is used in the calculation of the aerodynamic resistance. The height 2.5* hc is the minimum height. In case the height z is less than 2.5* hc, then the resistance of the roughness layer may be overestimated.

We assume that all the meteorological data are collected at the same height. However, for reanalysis data, in the case that wind speed is taken at 10m, air temperature at 2m, the more accurate way is to convert these measurements into the same height instead of setting z=2.5hc.

6. Table 2: Is there a relationship between Cab and Cs because senescenced leaves have lower Cab? Is there a relationship between Vcmax and Cab in terms of vertical variation? Why is Ball-Berry intercept parameter missed? Are their emissivity parameters?

Response: There are empirical relationships between Cab and Cs, Vcmax and Cab. As the reviewer expected, some studies reported an inverse relationship between Cab and Cs, and a positive linear relationship between Vcmax and Cab. However, these relationships are not universal, but vary with a number of factors, such as vegetation types. As a model designed for "all" plants, we have not introduced such empirical relationships in the model. It is our mistake for not including the Ball-Berry intercept parameter. We have added it to the table accordingly.

Yes, there are emissivity parameters for both leave and soil in the model. We have included the broadband thermal reflectance in the revised manuscript, which is 1-emissivity.

7. L218: I'm confused here. If we need to conduct a time series simulation or spatial simulation, do we need to provide variant tau and rho parameters?

Response: No, the users do not need to provide tau and rho parameters. They will be simulated by the vegetation model.

8. L227: While canopy FPAR can be obtained from outputs by FPAR = APAR/PAR, how can we get FPAR for leaves (sunlit/shaded at different layers)?

Response: Leaf FPAR is not specifically computed in the model, but it is linked with the absorptance of the leaf, which is 1-leaf reflectance –leaf transmittance. Strictly speaking, leaf FPAR is a spectrally integrated variable, and should be computed as APAR/PAR, where $APAR = \int_{400}^{700} E(\lambda)[1 - \rho(\lambda) - \tau(\lambda)]d\lambda$ and where $APAR = \int_{400}^{700} E(\lambda)d\lambda$.

9. Table 3: What's the relationship between LST, Tcave and Tsave? Is this LST term comparable to ground/satellite estimates?

Response: LST is determined by Tcave and Tsave. From the energy balance routine, we obtain the temperature of each individual leaf. Tcave represents the average temperature of all the leave. Similarly Tsave is the average temperature of sunlit and shaded soil. LST is computed from the Planck's law. Soil and leaf temperature, together with the net radiation, determine the canopy outgoing radiance, which is simulated with the radiative transfer models. From the outgoing radiance, we can estimate the black-body radiometric land surface temperature.

Yes, LST is comparable with remote sensing estimates with thermal sensors. For example, Duffour et al. (2015) compared the simulated LST with the measurements.

Duffour, C., Olioso, A., Demarty, J., Van der Tol, C., and Lagouarde, J.-P.: An evaluation of SCOPE: A tool to simulate the directional anisotropy of satellite-measured surface temperatures, Remote sensing of environment, 158, 362–375, 2015.

10. Section 3.3: How to input multi-layer vegetation parameters seems not mentioned. Also curious if vertical variation of meteorological data is modeled?

Response: We have added that "In comparison with the original SCOPE, SCOPE 2.0 accepts vertical profiles of leaf properties (such as chlorophyll content) as inputs. If single values of the Fluspect parameters in Table 2 are provided, the model will assume the canopy is vertically homogeneous. "

11. Figure 5: This figure is not cited in the text.

Response: Thank you for sorting this out. We have cited this figure in the text in section 3.3.

12. Figure 8: Does the bias indicate that the lite option is not suitable for thermal remote sensing? I think such clarification might be useful to users.

Response: We agree that such clarification is needed. The Figure shows that the difference in TOC SIF is around 0.1 Wm-2um-1sr-1, and around 1 degree in the surface temperature simulation. However, the applicability of the lite option depends on specific purposes and the desired accuracy.

13. L418. While the "improved computational efficiency" is shown in Table 4, the "improved model stability" does not have evidence in the manuscript.

Response: We have solved a few bugs in the code in the past 11 years, which led to a more stable model.

14. L419. The topic "understory and overstory" is never mentioned in the manuscript. Does SCOPE 2.0 has understory and overstory LAI separated?

Response: Canopies with understory and overstory are considered as a two-layer canopy. This can be simulated with SCOPE 2.0. We have introduce the idea of understory and overstory in section 3.3 as follows:

"In reality heterogeneity of leaves within a vegetation canopy might be infinitely large and cannot be specified in a model. This requires a simplification of the canopy in the model, and the use of two- or three-layer representation is the most common way. For example, forests usually have understory and overstory, and crops at the senescent stage have two or three distinctive layers with brown or green leaves.

---

## Author Comment (AC3) · 25 Apr 2021

It should be noted that to compute GPP from the simulated photosynthesis, it is necessary to add the simulated dark respiration: GPP = Ac + Rdparam\*Vcmax25\*LAI. In other words, for actual GPP simulation leaf respiration should be disabled, setting Rdparam to 0. The difference with some other canopy-scale models is that SCOPE applies the FvCB model at leaf level (disaggregated), whereas many other models apply this model at aggregated (big leaf or sun-shade model) scales. Our hypothesis is that this results in higher required Vcmax25 values on SCOPE compared to other models. This is an important topic, which requires a detailed investigation.

---

## Author Comment (AC4) · 25 Apr 2021

1. Response to the comment on typical diurnal cycles of G

The (negative) night time Rn and G (shown in the Figure in our previous response) seems to be underestimated (in absolute value) compared to what has been reported in other studies (e.g. Van der Tol, 2012). We hypothesize that this is at least partly due to the turbid medium representation of the vegetation, which may lead to underestimation of the gap fraction (and thus the exposed part of the soil) and thus the night-time radiative cooling of the soil.

2. Response to the comment on FPAR of the leaves

The model outputs the APAR for sunlit and shaded leaves (per layer), by computing the spectral integration of the product of (leaf) irradiance and absorptance of the leaf, which is 1-leaf reflectance –leaf transmittance. Because the model differentiates leaves of different orientation (and exposure to the Sun) this is done for all leaf elements. The APAR for all the sunlit or shaded leaves combined is calculated by integrating the product of the individual leaf contributions and their probability of occurrence, which is determined by the leaf orientation distribution and the canopy gap fraction. Finally, the FPAR can be calculated by the user from the APAR by dividing FPAR by the incident PAR, which is also the output of the model, but for the canopy as a whole. The FPAR is for the sunlit and shaded fractions separately, in APAR/iPAR_leaf, is not output.

3. Response to the comment on the relationship between LST, Tcave and Tsave

From the energy balance routine, we obtain the temperature of each individual leaf, which is the equilibrium temperature at which the energy balance closes (radiation, sensible, latent and ground heat fluxes). Tcave represents the average temperature of all the leaves. Similarly, Tsave is the average temperature of sunlit and shaded soil. This is a simple arithmetic average, which is strictly not physically sound, but it is nevertheless a good indicator. LST is computed from Planck's law once the equilibrium soil and leaf temperature are known. First, the outgoing radiance in the observation direction is simulated with the thermal radiative transfer model. This simulation is carried out twice: - Once for thermally black soil and leaves (Lob) - Once with the actual emissivities of soil and leaves (Lo). The whole-stand effective emissivity is then calculated as: Emissivity = Lo/Lob which holds a value between the soil and leaf emissivity. The LST is then estimated by inversion of the Stefan-Boltzman equation from Lo and the emissivity. This LST is comparable to radiometric observations of temperature from proximal or remote sensing. For example, Duffour et al. (2015) compared the simulated LST with the measurements. Duffour, C., Olioso, A., Demarty, J., Van der Tol, C., and Lagouarde, J.-P.: An evaluation of SCOPE: A tool to simulate the directional anisotropy of satellite-measured surface temperatures, Remote sensing of

environment, 158, 362–375, 2015.

4. Response to the comment on bias in LST simulations in Figure 8

The figure shows that the difference in TOC SIF is around 0.1 Wm-2um-1sr-1, and around 1 degree in the surface temperature simulation. Thus the difference in radiance is minimal, while the difference in average temperature is relatively higher (compared to the natural spatio-temporal variability). This is not an error, but simply due to the non-linear relation between temperature and irradiance in the Planck law (see our response to the point of average temperature vs LST). However, the applicability of the lite option depends on specific purposes and the desired accuracy.

---

## Author Response (AR1)

SCOPE 2.0: A model to simulate vegetated land surface fluxes and satellite signals

**Response to reviewers' comments**

Peiqi Yang, Egor Prikaziuk, Wout Verhoef, Christiaan van der Tol\*

**A list of all relevant changes**

- adding some introduction of other models
- specifying some model limitations, such as clumping effects, withincanopy variation of meteorological conditions
- mentioning the difference between SCOPE2.0 and other land surface models for GPP simulations
- including a more comprehensive guide on the use of multi-layer option in SCOPE 2.0
- discussing the future directions for model developments.

**Yulu Zeng**

**General comments**

Thanks for inviting me to review this manuscript. Earlier versions of SCOPE have been widely used in SIF, GPP, energy balance and thermal signal simulations. This paper mainly summarized the recent progress of the revised version, SCOPE 2.0, which includes (1) the consideration of multi-layer vertical variation of chlorophyll content and leaf angle distributions; (2) the adoption of the BSM soil reflectance model to account for the soil moisture; (3) the impact of the xanthophyll cycle on the leaf-canopy reflectance and (4) speed acceleration optimization. I believe these advances are of interest to the remote sensing and ecology modeling communities, and thus this paper matches the scope of the GMD journal very well.

Overall, this paper is well written and structured. I have a few comments from the perspective of a SCOPE user, and the authors may choose to consider or not according to the long-term plan of SCOPE improvements and the amount of effort needed. Dear Dr. Zeng, We thank you for the positive and encouraging feedback. We studied your comments with attention and revised our manuscript accordingly. Your comments are constructive and helpful, and we provide itemized responses to them below.

1. P21, L390: Currently in SCOPE2.0, atmospheric properties are the input parameters that determine the proportion of the direct and diffuse solar radiation. As the authors mentioned in Line 390, for the simulation of a specific site (e.g., some Fluxnet and PhenoCam sites), PAR and diffuse PAR ratio are usually available instead of the atmospheric properties. This makes the simulation of SIF and photosynthesis to be difficult at the diurnal or seasonal cycle with different diffuse PAR ratio. It would be more convenient for users in ecology community, if PAR and diffuse PAR ratio could be used as input parameters in SIF and photosynthesis simulations.

Response: Indeed, in many cases, the atmospheric properties or the incoming irradiance spectra are not available. The suggested option to provide the diffuse: direction ratio as input would then be convenient for the user. The reason why this option is absent is that the ratio varies with wavelength (diffuse radiation varies from blue-ish skylight to white light reflected by clouds, and direct radiation varies from white to reddish depending on the solar angle). Thus the whole spectrum of the diffuse and direct radiation would be needed.

As an alternative to providing atmospheric properties, SCOPE 2.0 offers the option to provide direct and diffuse irradiance spectra as input. If the user has measurements of the direct: diffuse ratio, then the corresponding spectra could be estimated and provided as input, by making some assumptions about the spectral distribution of the ratio.

2. Second is about the validation. As far as I know, there is rare literature about the validation of SCOPE over high productivity areas with GPP > 40 umol CO2 m-2 s-1. From my experience of using SCOPE to simulate GPP of soybean at the Corn Belt in the US and in summer, it is very difficult to be able to achieve the GPP simulations to be larger than 40 umol CO2 m-2 s-1 with the field PAR, temperature, chlorophyll content, etc measurements, unless we set the unmeasured Vcmax to be larger than 200 umolm-2 s-1. Of course, this is unreasonable. Validation of SCOPE simulated GPP over high productivity areas would give more confidence and guidance to the SCOPE users in ecology community.

Response: This is an interesting point. We have checked this issue and confirm that the model indeed requires very high Vcmax to simulate GPP>40 umol  $CO_2$  m-2 s-1 for C3 plants, such as soybean.

It should be noted that to compute GPP from the simulated photosynthesis, it is necessary to add the simulated dark respiration:

GPP = Ac + Rdparam\*Vcmax25\*LAI.

We will explain this is in the revision. Nevertheless, we have conducted several numerical experiments with different model parameterization and conclude that the required Vcmax25 values are still high. According to our investigation, we conclude that this is most likely inherited from the FvCB photosynthesis model. The difference with some other canopy-scale models is that SCOPE applies the FvCB model at leaf level (disaggregated), whereas many other models apply this model at aggregated (big leaf or sun-shade model) scales. Our hypothesis is that this results in higher required Vcmax25 values on SCOPE compared to other models. This is an important topic, which requires a detailed investigation. Therefore, we don't want to rush in drawing any conclusion, or propose any solution in the current manuscript before a fundamental study on the photosynthesis model is done.

3. P 15, Fig. 4: For the xanthophyll cycle, Vilfan et al (2018) only focused on the leaf scale. In fact, there are already many canopy-scale PRI field observations acquired during the plant stress in recent studies. Showing the capability (and good performance) of SCOPE2.0 with field data to capture the plant stress by quick response (and accurate simulation) of PRI (or CCI) would be more interesting and convincing than model comparisons if possible.

Response: Yes, validating the model with field measurements of PRI or xanthophyll cycle induced reflectance variation is a necessary experiment to show the accuracy of the model. The challenge is the accurate parameterization of the model. Since we aim to capture small changes in reflectance from 520 to 560 nm, the inputs of the model variables have to be accurately provided. Although there are numerous data sets measuring canopy-scale PRI at diurnal and seasonal scale, the associated leaf biochemical and canopy structure measurements might not be sufficient to reproduce the PRI variations.

We are working on this subject, but we think it requires many details, which is too much to be incorporated in the current manuscript in which the model is presented with focus on the technical (implementation) aspects.

4. P17, L 345-350: Usually it is difficult to determine how many layers should be set in SCOPE for a specific vegetation species, e.g., corn with narrow and long inclined leaves, or taro with big leaves but not many layers. In my understanding, the layers in SCOPE and the leaf layer in reality are different. For example, if there is only one flat big leaf over the ground, sensors can always observe the hot-spot effect in all viewing directions, while this kind of situations are difficult to consider in model settings. Of course, this is the gap between abstract models and nature in reality. The current model is correct under general assumptions of radiative transfer modelling, while caveats and more guidance may be still needed for users to correctly use the model and achieve accurate simulations.

Response: Yes, the reviewer is totally right on the difference between the numerical differentiation into layers in SCOPE and leaf-layers in a vegetation canopy. SCOPE, as well as SAIL, assumes the canopy is a turbid medium. The layers are needed for numerical discretization.

We also agree that more guidance on the setting of SCOPE 2.0 for multilayer canopies is needed. It is possible to parameterize vertical heterogeneity in the vegetation canopy. We will revise the text with the following addition:

"The true heterogeneity of leaves within a vegetation canopy may be too large to fully implement in the model. Thus, a simplification of the canopy may be needed: Two- or three-layer representations are most common. However, it is noted that more layers are possible in SCOPE 2.0 for specific purposes as illustrated in Yang et al. (2017)".

The hotspot effects have been included by correcting the directional radiance for the effects of finite size of leaves. The parameter that is used is the ratio between leaf width and vegetation height. Otherwise, the dimensions and shapes of leaves are not used in the model.

5. P16, Fig. 5: Does SCOPE2.0 have the capability to simulate the scene with different leaf sizes at different layers? Even if the leaf size is the same for all layers, the hot spot factor or leaf specific dimension could vertically vary with

different multi-layer leaf angle distributions (Kuusk 1991). Seems this issue was not considered and the hot-spot effect was not evaluated or discussed in mSCOPE and SCOPE2.0 manuscripts. Not quite sure how large are the uncertainties by this issue to the reflectance around the hot spot directions. If fixed as one value, I suppose the multi-layer hot spot factor could be closer to the hot spot factor of the upper layer instead of the vertically averaged value. Maybe uncertainties by this issue could be evaluated by 3D multi-layer simulations.

Kuusk, A. (1991). The hot spot effect in plant canopy reflectance. In Photon-Vegetation Interactions (pp. 139-159). Springer, Berlin, Heidelberg.

Response: Yes, it is possible, but the option is not provided in the model input. In the mSCOPE model, this has not been considered, and we assumed the hotspot implementation in mSCOPE remains correct as it is identical to the SAIL model.

As the reviewer mentioned, the leaf size would only affect reflectance/radiance simulation in the hotspot direction in SCOPE 2.0. We feel it is an advanced use of the multi-layer radiative transfer modelling. For the general use of the model, this might not be a major issue, and introduction of the vertical variation of the leaf size could be a distraction to users. However, we would be happy to discuss the potential of this option.

6. P12, L230: The BSM model which simulates the isotropic soil reflectance was adopted in this study. For sparse vegetation canopies such as shrubland with low fractional vegetation cover and considerable soil roughness, the soil anisotropy and hot-spot effects are also important to the canopy reflectance for the chlorophyll content and leaf area index retrievals. Hope in the future the soil anisotropic model, e.g., the Hapke model, could be incorporated in the SCOPE framework at least for the soil single scattering contribution.

Response: Thank you for the suggestion. We totally agree that the importance of the hotspot effects of soil reflectance, which is not considered the BSM model. The combination of the Hapke and BSM models could be an improvement. We will consider this as an improvement of the SCOPE 2.0 model.

7. A discussion paragraph or section maybe needed to show the future directions of SCOPE improvements. Recently the leaf specular reflection has been reported to considerably contribute to the canopy reflectance especially over needle leaf

forest, while this effect seems was not well considered in the current version. Besides, the 3D complex forest structures which can cast crown-scale dark shadows may also be the challenge for the current 1D models.

Response: Thank you for the suggestion. We have decided to add several sentences at the end of the conclusion about the future directions.

With the aim for accurate simulations of vegetative land surface processes and remote sensing signals, the models are constantly improved. Some important features, such as canopy clumping effects, crop yield simulation, leaf specular reflection and soil BRDF effects, are considered as the future directions of SCOPE improvements.

8. Seems the canopy coverage Cv was considered in the code of SCOPE2.0, while was not mentioned in the current manuscript.

Response: Cv was introduced with the intention to include the clumping effects. However, we find the effects of clumping is rather complicated and thus we have removed this parameter in the code. Our plan is to include the clumping effects in the future.

9. Congratulations to the authors for the several important advances of SCOPE2.0, and I can foresee that this paper as a milestone of SCOPE will have considerable impact on the remote sensing and SIF community. Not all my concerns need to be addressed this time according to the feasibility and available dataset, and some of them could be in the discussion. The accurate description and guidance of the model can better meet the users' needs and expectations. The well validation of the model by field observations can help to bridge the gap between abstract models (modeler community) and complex reality (user community with observations), and is also helpful for the future model improvements.

Response: Thank you again for your comments. We totally agree with your points as model developers. We aim at providing a tool for the community to better understand the real world. We have revised the manuscript by accounting for most of the comments from you and the reviewer #2. We hope our revision and response have addressed your concerns.

**Anonymous Referee #2**

**General comments**

This study improves the widely-used radiative transfer and biophysical model SCOPE by implementing 1) soil reflectance simulation, 2) xanthophyll cycle modulation, 3) vertical variations of vertical properties, 4) dynamic ground heat flux simulation, 5) a full energy balance closure solution, and 6) multiple strategies for computational efficiency. These improvements are significant advances and I believe the proposed SCOPE 2.0 will benefit the vegetation remote sensing community. The paper is well written and I only one major comment followed by several minor comments.

Dear reviewer, we thank you for your precious time as well as your constructive and positive comments.

**Major comments**

1. While the improved algorithms are well described, the performance/effects of the new algorithms are not fully demonstrated.

(1) Can you compare TOC reflectance, GPP and SIF between using a vertically variant Cab and using an invariant Cab? This is very interesting as I see many studies, including ESA's products, interpret canopy chlorophyll content as the product of LAI and Cab without considering the vertical variation of Cab. SCOPE 2.0 can help us understand this impact.

Response: We agree with the reviewer on the importance of vertical variation of leaf biophysical properties. Although the specific experiment/comparison is not included in this manuscript, we have already done this in the other paper about the mSCOPE model, which is a branch of the SCOPE model that includes vertical heterogeneity. In that work, we show the effects of vertical profiles of Cab on TOC reflectance, GPP, SIF and light use efficiency. To avoid repetition, we decided not to include such comparison in this manuscript, but added sentences clearly stating the comparison is available in the mSCOPE paper.

Yang, P., Verhoef, W., & van der Tol, C. (2017). The mSCOPE model: A simple adaptation to the SCOPE model to describe reflectance, fluorescence and photosynthesis of vertically heterogeneous canopies. Remote sensing of environment, 201, 1-11.

(2) Can you compare typical diurnal cycles of G between G = 0.35Rn and the new parameterization?

Response: The (negative) night time Rn and G seems to be underestimated (in absolute value) compared to what has been reported in other studies (e.g. Van der Tol, 2012). We hypothesize that this is at least partly due to the turbid medium representation of the vegetation, which may lead to underestimation of the gap fraction (and thus the exposed part of the soil) and thus the night-time radiative cooling of the soil.

(3) Can you use figures/tables to show

1) how energy balance closure is improved by using the new iteration algorithm,

Response: The older version 1.74 requires on average 59 iterations to close the energy balance, the 2.0 version requires on average 9. Furthermore, the standard deviations are dramatically different 59+/-92 vs 9+/-9.

2) why Eq. 7 is a sufficiently accurate approximation?

Response: Equation 7 is a linearization of the relation between temperature and energy balance error. This linearization is estimated analytically, which is much faster than calculating the derivative numerically. It is sufficient because it provides the slope, and thus a good update step of the temperature. The update is not exact, because the net radiation of the leaves also depends on the temperature of the other leaves (through the radiative transfer model). This cannot be solved analytically. The estimate is still sufficient, indicated by the quick convergence of the energy balance. The number of iteration steps in SCOPE 2.0 is significantly lower than in earlier versions of the model.

$$\delta e_{bal} = Rn - \lambda E - H \tag{1}$$

$$\frac{\delta Rn}{\delta T} = \frac{\delta Rn_{sw}}{\delta T} + \frac{\delta Rn_{lw}}{\delta T} = 0 - \frac{\delta \sigma_{SB} * (T + 273.15)^4}{\delta T} = -4 * \sigma_{SB} * (T + 273.15)^3$$
(2)

$$\frac{\delta H}{\delta T} = \frac{\delta \rho * c_p./r_a * (T_c - T_a)}{\delta T} = \rho * c_p./r_a$$
(3)

(5)

$$\frac{\delta\lambda E}{\delta T} = \frac{\delta\rho/(r_a + r_s) * \lambda * (q_i - q_a)}{\delta T} = \rho/(r_a + r_s) * \lambda * \frac{\delta q_i}{\delta T}$$
(4)

$$q_i = 6.107 * 10^{\frac{7.5 * T}{237.3 + T}}$$

$$\frac{\delta q_i}{\delta T} = s = q_i * \ln(10) * \frac{7.5 * 237.3}{(237.3 + T)^2}$$
(6)

**Minor comments:**

1. L16: I would suggest add some introduction of other models that can simulate radiative transfer and fluxes and provide distinct feature of SCOPE comparing to these models.

Response: We will do so in the revision (e.g. ACACIA, CUPID, SiB)

2. L87: SCOPE lacks the consideration of clumping effect, right? If so, I suggest add some words about that so that users can keep it in mind.

Response: Yes, it is right, although we are working on including this effect. As suggested, we have added a sentence stating the lack of clumping effects in SCOPE2.0.

3. Table 1: The term "each leaf" is unclear. How many "leaves" in SCOPE 2.0? 13\*36\*n for sunlit and shaded, respectively?

Response: We see the confusion here. To make it clearer, we changed "each leaf" to "individual leaves".

The model differentiates 13\*36\*n sunlit leaves and n shaded leaves (where n is the number of layers). The leaves are different from each other by their orientation and leaf biophysical properties. In total, 13\*36 types of leaf orientations are defined in SCOPE 2.0, and the biophysical properties of the leaves in the n vegetation layers can vary among the layers.

4. L128: What type of aerodynamic resistance scheme is used in SCOPE 2.0? Series or parallel?

Response: Parallel. The leaves and soil are all parallel sources, so the model is a multi-source model. Each leaf and soil element has three resistances in series: stomatal/soil surface, leaf/soil laminar boundary, within vegetation, and above the vegetation. The the resistance above the vegetation is has an equal value for all leaves and soil. The within-canopy and boundary layer differs between soil and leaves, and the surface/stomatal resistance is different for all individual leaf classes (described in more detail in Van der Tol et al. (2009) and Wallance and Verhoef (1997)). We will add a few lines of explanation in the revision.

5. L180: Why is z "typically 2.5 times the vegetation height"? If we use meteorological data from site data or reanalysis data, they are fixed, right?

Response: The height should indeed be set to the height of the meteorological tower, and this height is used in the calculation of the aerodynamic resistance. The height 2.5xz is the minimum height. The resistance scheme assumes that at above this height (2.5z), wind profile is logarithmic. If z is less than this, then the resistance of the roughness layer may be overestimated.

We assume that all the meteorological data are collected at the same height. However, for reanalysis data, in the case that wind speed is taken at 10m, air temperature at 2m, the more accurate way is to convert these measurements into the same height before providing them to the model instead of setting z=2.5 hc.

6. Table 2: Is there a relationship between Cab and Cs because senescenced leaves have lower Cab? Is there a relationship between Vcmax and Cab in terms

of vertical variation? Why is Ball-Berry intercept parameter missed? Are their emissivity parameters?

Response: All these parameters can be set independently, in order to allow flexibility of the model. However, the user can use empirical relationships between Cab and Cs, Vcmax and Cab or Cw before providing these data as input.

As the reviewer mentions, some studies reported an inverse relationship between Cab and Cs, and a positive linear relationship between Vcmax and Cab. However, these relationships are not universal, but vary with a number of factors, such as vegetation types. As a model designed for "all" plants, we have not introduced such empirical relationships in the model. It is our mistake for not including the Ball-Berry intercept parameter. It is part of the model and we have added it to the table accordingly.

Yes, there are emissivity parameters for both leaves and soil in the model. The model uses broadband thermal reflectance and transmittance for leaves, and thermal reflectance for the soil as user defined input. These are related to the emissivity via Kirchhoff's law (emissivity = 1-reflectance-transmittance). We will include this in the manuscript, together with the simulation of whole-stand effective emissivity and land surface temperature.

7. L218: I'm confused here. If we need to conduct a time series simulation or spatial simulation, do we need to provide variant tau and rho parameters?

Response: No, the users do not need to provide tau and rho parameters. They will be simulated by the vegetation model. The users can provide the (varying) pigments and soil properties that are the input to the rho and tau simulation.

8. L227: While canopy FPAR can be obtained from outputs by FPAR = APAR/PAR, how can we get FPAR for leaves (sunlit/shaded at different layers)?

Response: The model outputs the APAR for sunlit and shaded leaves (per layer), by computing the spectral integration of the product of (leaf) irradiance and absorptance of the leaf, which is 1-leaf reflectance –leaf transmittance. Because the model differentiates leaves of different orientation (and exposure to the Sun) this is done for all leaf elements. The APAR for all the sunlit or shaded leaves combined, is calculated by integrating the product of the individual leaf

contributions and their probability of occurrence, which is determined by the leaf orientation distribution and the canopy gap fraction.

Finally, the FPAR can be calculated by the user from the APAR by dividing FPAR by the incident PAR, which is also output of the model, but for the canopy as a whole. The FPAR is for the sunlit and shaded fractions saparately, in APAR/iPAR\_leaf, is not output.

9. Table 3: What's the relationship between LST, Tcave and Tsave? Is this LST term comparable to ground/satellite estimates?

Response: From the energy balance routine, we obtain the temperature of each individual leaf, which is the equilibrium temperature at which the energy balance closes (radiation, sensible, latent and ground heat fluxes). Tcave represents the average temperature of all the leaves. Similarly Tsave is the average temperature of sunlit and shaded soil. This is a simple arithmetic average, which is strictly not physically sound, but it is nevertheless a good indicator.

LST is computed from the Planck's law once the equilibrium soil and leaf temperature are known. First the outgoing radiance in observation direction is simulated with the thermal radiative transfer model. This simulation is carried out twice:

- Once for thermally black soil and leaves (Lob)
- Once with the actual emissivities of soil and leaves (Lo).

The whole-stand effective emissivity is then calculated as:

Emissivity = Lo/Lob

which holds a value between the soil and leaf emissivity.

The LST is then estimated by inversion of the Stefan-Boltzman equation from Lo and the emissivity. This LST is comparable to radiometric observations of temperature from proximal or remote sensing. For example, Duffour et al. (2015) compared the simulated LST with the measurements.

Duffour, C., Olioso, A., Demarty, J., Van der Tol, C., and Lagouarde, J.-P.: An evaluation of SCOPE: A tool to simulate the directional anisotropy of satellitemeasured surface temperatures, Remote sensing of environment, 158, 362–375, 2015.

10. Section 3.3: How to input multi-layer vegetation parameters seems not mentioned. Also curious if vertical variation of meteorological data is modeled?

Response: We have added that "In comparison with the original SCOPE, SCOPE 2.0 accepts vertical profiles of leaf properties (such as chlorophyll content) as inputs. This is done via a table, in which optical properties can be specified for user defined LAI intervals. If single values of the Fluspect parameters in Table 2 are provided, the model will assume the canopy is vertically homogeneous.

Vertical variation of meteorological data is not fully simulated, only the levels: above the roughness layer, in the vegetation layer, in the leaf boundary layer are differentiated.

11. Figure 5: This figure is not cited in the text.

Response: Thank you for pointing this out. We have cited this figure in the text in section 3.3.

12. Figure 8: Does the bias indicate that the lite option is not suitable for thermal remote sensing? I think such clarification might be useful to users.

Response: We agree that such clarification is needed. The figure shows that the difference in TOC SIF is around 0.1 Wm-2um-1sr-1, and around 1 degree in the surface temperature simulation. Thus the difference in radiance is minimal, while the difference in average temperature is relatively higher (compared to the natural spatio-temporal variability). This is not an error, but simply due to the non-linear relation between temperature and irradiance in the Planck law (see our response to the point of average temperature vs LST). However, the applicability of the lite option depends on specific purposes and the desired accuracy.

13. L418. While the "improved computational efficiency" is shown in Table 4, the "improved model stability" does not have evidence in the manuscript.

Response: We have solved a few bugs in the code in the past 11 years and improved the update step in the iteration, which led to a more stable model in

terms of energy balance closure success rates. We will add a column with the percentage of the cases in which the energy balance closed for all elements.

14. L419. The topic "understory and overstory" is never mentioned in the manuscript. Does SCOPE 2.0 has understory and overstory LAI separated?

Response: Canopies with understory and overstory are considered as a twolayer canopy. This can be simulated with SCOPE 2.0. We have introduced the idea of understory and overstory in section 3.3 as follows:

"In reality heterogeneity of leaves within a vegetation canopy might be infinitely large and cannot be specified in a model. This requires a simplification of the canopy in the model, and the use of two- or three-layer representation is the most common way. For example, forests usually have understory and overstory, and crops at the senescent stage have two or three distinctive layers with brown or green leaves.